# RECURRENT OFF-POLICY DEEP REINFORCEMENT LEARNING DOESN'T HAVE TO BE SLOW

## ABSTRACT

Recurrent off-policy deep reinforcement learning models achieve state-of-the-art performance but are often sidelined due to their high computational demands. In response, we introduce RISE (Recurrent Integration via Simplified Encodings), a novel approach that can leverage recurrent networks in any image-based off-policy RL setting without significant computational overheads via using both learnable and non-learnable encoder layers. When integrating RISE into leading non-recurrent off-policy RL algorithms, we observe a 35.6% human-normalized interquartile mean (IQM) performance improvement across the Atari benchmark. We analyze various implementation strategies to highlight the versatility and potential of our proposed framework. We make our code available in the supplementary material.

## 1 INTRODUCTION

Deep Reinforcement Learning (RL) has achieved many successes, such as playing Dota 2 (Berner et al., 2019), controlling nuclear fusion plasma (Degrave et al., 2022), and solving mathematical problems (Trinh et al., 2024). Despite this, widespread adoption of RL has been slow, largely due to the high computational costs, making it inaccessible to almost all researchers, developers, and businesses (Ceron and Castro, 2021). Therefore, replicating RL's success with widely available compute in a reasonable amount of walltime remains an outstanding and critical problem for the field.

One notable feature of many state-of-the-art RL algorithms (Badia et al., 2020; Hafner et al., 2023) is the use of recurrent models, as it can overcome partial observability and improve long-term credit assignment. For on-policy RL, integrating recurrent models has minimal computational cost; however, the performance of on-policy algorithms such as PPO (Schulman et al., 2017) and PQN (Gallici et al., 2024) consistently falls short of their off-policy counterparts such as MEME (Kapturowski et al., 2023) and Beyond The Rainbow (BTR, Clark et al. (2024)).

While recurrent models have been integrated into off-policy algorithms, substantially boosting their performance, this comes with significant compute and walltime costs. As a result, off-policy recurrent RL algorithms have been inaccessible to a majority of RL researchers. The root cause of this is the need to encode a long sequence of observations (using expensive convolutional layers in image-based environments) to provide the context when computing a Q-value during gradient updates.

To mitigate this, prior algorithms such as R2D2 (Kapturowski et al., 2018) often learn from sequential trajectories of observations, rather than single transitions. While this wastes fewer encoder passes per updated Q-value, it comes with a myriad of disadvantages (discussed in Section 3), such as temporally correlated updates and needing to use extremely large batches, which is inefficient in RL (Obando Ceron et al., 2023). Therefore, those with fewer compute resources are still unable to utilize recurrent models in off-policy RL.

Therefore, we introduce RISE (Recurrent Integration via Simplified Encodings) to efficiently integrate recurrent neural networks into off-policy RL, while achieving similar results to prior work (Figure 1). In Section 3, we propose a dual-stream architecture where recent observations are processed through learnable encoders (as in standard approaches), while the long context window is encoded via a non-learnable encoding, such as pretrained networks, and fed to a recurrent model. This separation allows the recurrent component to leverage long-term context without requiring expensive recomputation of encodings during training, as the non-learnable encodings can be precomputed and stored in the replay buffer (Lin, 1992). RISE's architecture is visualized in Figure 2.

In summary, the key contributions of our work are:

- We propose a general framework, 'Recurrent Integration via Simplified encodings' (RISE) that uses non-learnable encodings (such as pretrained models) as inputs to a recurrent layer for efficient integration of recurrent models in off-policy RL.

- We demonstrate performance improvements on the Atari 2600 (Bellemare et al., 2013; Towers et al., 2024), Procgen (Cobbe et al., 2020), Vizdoom (Kempka et al., 2016) and Miniworld (Chevalier-Boisvert et al., 2023) benchmarks using our framework.

- We provide a detailed analysis of the various ways our framework can be implemented, including context length, recurrent paradigm, and the value of pre-trained vision encodings.

- When our framework is applied to Beyond The Rainbow (BTR) (Clark et al., 2024), to our knowledge, this produces the highest performance algorithm capable of running on a single high-end desktop PC within a day of walltime.

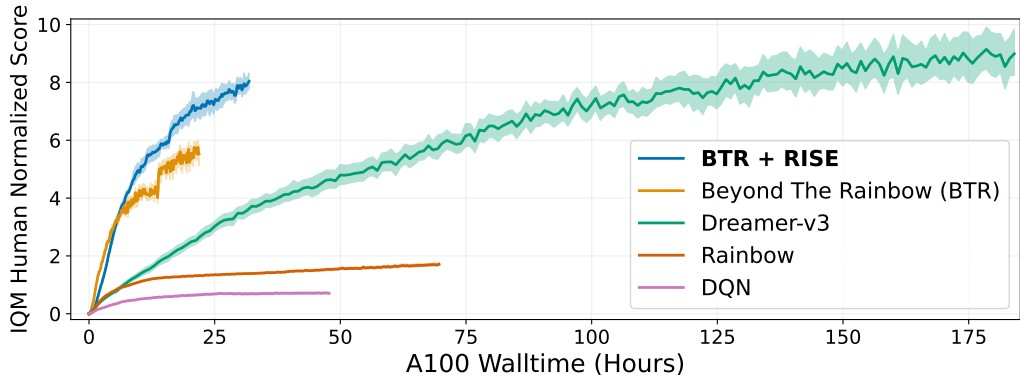

Figure 1: Interquartile mean human-normalized performance showing the BTR algorithm with and without our proposed method, Recurrent Integration via Simplified Encodings (RISE), on the Atari-55 benchmark. All algorithms use 200 million frames, and shaded areas show 95% bootstrapped confidence intervals. Dreamer-v3, Rainbow and DQN refer to (Hafner et al., 2023; Hessel et al., 2018; Mnih et al., 2015) respectively.

## 2 BACKGROUND

**Reinforcement Learning -** We consider the typical formulation of discounted infinite-horizon RL in Markov Decision Processes (MDP) (Puterman, 2014); this is defined by the tuple $(\mathcal{S}, \mathcal{A}, \mathcal{P}, \mathcal{R})$, where $\mathcal{S}$ is the set of states, $\mathcal{A}$ is the set of actions, $\mathcal{P} : \mathcal{S} \times \mathcal{A} \rightarrow \Delta(\mathcal{S})$ is the stochastic transition function, and $\mathcal{R} : \mathcal{S} \times \mathcal{A} \rightarrow \mathbb{R}$ is the reward function. The objective is to obtain a policy $\pi : S \rightarrow \Delta(\mathcal{A})$ that maximizes the expected sum of discounted rewards $\mathbb{E}_\pi[\sum_{t=0}^{\infty} \gamma^t r(s_t, a_t)]$, where $\gamma \in [0, 1)$ is the discount rate. One popular family of methods is Q-Learning (Watkins and Dayan, 1992), whereby a function $Q^\pi(s, a) = \mathbb{E}_\pi[\sum_{t \geq 0} \gamma^t r_t | s_t = s, a_t = a]$ represents the expected discounted reward achieved by taking action $a_t$ in state $s_t$ and subsequently following the policy $\pi$ which can be derived from the Q-function using the $\epsilon$-greedy operator $\mathcal{G}_\epsilon$ (Sutton, 2018). To allow state-action generalization, Mnih (2013) used a deep neural network with parameters $\theta$ to estimate the action-value function, $Q^\pi(s, a : \theta)$. Following this method, numerous methods would be introduced to stabilize (Van Hasselt et al., 2016; Mnih et al., 2015) and improve (Fortunato et al., 2018; Wang et al., 2016; Bellemare et al., 2017) the performance of such networks, many of which would later be combined into a single algorithm, Rainbow DQN(Hessel et al., 2018). Beyond The Rainbow (BTR) would later adapt this work, increasing performance and walltime efficiency by adding more components.

**Transition-Based off-policy RL -** One practical defining feature of the aforementioned algorithms is that they collect and store transitions, defined as $(s_t, a_t, r_t, s_{t+1}, \perp_t)$, where $\perp$ is a boolean value representing whether a terminal state has been reached. Transitions are stored in a Replay Buffer (Lin, 1992; Schaul, 2015), and fetched to perform gradient updates with target $r_t + \gamma(\neg \perp_t) \max_{a \in A} Q_{\theta'}(s_{t+1}, a)$. This method allows the agent to learn from prior experiences

generated by different policies (referred to as the behavior policy, $\mu$). While maintaining a Replay Buffer is typically computationally slower than on-policy methods (Schulman et al., 2017), they are considerably faster than trajectory-based off-policy methods which learn from trajectories $(s_t, a_t, r_t, \perp_t, ..., s_{t+m}, a_{t+m}, r_{t+m}, \perp_{t+m})$ for some length $m$.

**Recurrent Models -** RL has followed much of Deep Learning in using recurrent models to handle long sequences of data. For on-policy algorithms, this is a trivial extension; however, off-policy methods struggle to utilize these methods without substantial additional computational cost. To form a Markov state $s_t$, recurrent models need to pass all previous observations through an encoder function, $s_t = f(o_1, o_2, \cdots, o_t)$. Since this can be infeasible for long episodes, R2D2 (Kapturowski et al., 2018) explored cheaper methods without this limitation for the trajectory-based setting, proposing *burn-in* and *stored-state*. Burn-in would encode $l$ observations before those being updated, with $s_t = f(o_{t-l}, o_{t-l+1}, \cdots, o_t)$; however, this wastes many encoder passes on observations that don't receive updates. Stored-state saved the model's outputs (such as a hidden state) when generated, such that they could be fetched from the Replay Buffer. However, this suffers from state-staleness, where the stored value will become progressively more inaccurate as the model updates its parameters, $\theta$. Critically, no previous image-based recurrent models have removed the requirement of numerous convolutional network passes during training steps.

## 3  COMPUTATIONALLY EFFICIENT RECURRENT MODELS

When using recurrent models (such as LSTMs) in off-policy RL, the major computational burden comes from needing to pass all observations in the context window through layers preceding the LSTM, which, for image-based tasks are expensive convolutional neural networks (CNNs). We propose Recurrent Integration via Simplified Encodings (RISE) to bypass this restriction by using and storing fixed observation encodings. This elegantly avoids the need to recompute encodings, as if the inputs to the LSTM is non-learnable (does not change throughout training), these inputs can be computed once, stored and fetched quickly when needed. While the framework presented in this paper applies to any off-policy RL setting, including Deep Deterministic Policy Gradients (DDPG, Lillicrap (2015)), Soft Actor Critic (SAC, Haarnoja et al. (2018)), and Twin Delayed DDPG (TD3, Fujimoto et al. (2018)), we primarily consider the application of our framework to Rainbow DQN (Hessel et al., 2018) and Beyond The Rainbow (BTR, Clark et al. (2024)).

More formally, let $\phi$ represent the encoder function (most commonly convolutional layers), which takes the observation, let $\psi$ represent the LSTM layer, which takes a sequence of embeddings (such as those produced by $\phi$), and let $\omega$ represent the linear layers, which also take an embedding. In non-recurrent RL, the model's output $Q(s_t, a_t)$ is simply $\omega(\phi(o_t))$. In R2D2-style recurrent RL systems, $Q(s_t, a_t) = \omega(\psi(\phi(o_{t-l}), \phi(o_{t-l+1}), ..., \phi(o_t))))$, requiring every observation in the context window to pass through $\phi$. In RISE, let us introduce an additional function $\varphi_{\cancel{\theta}}$ that is not learnable and takes $o_t$ to produce an embedding, and $\Omega$, a linear layer used to project $\psi$'s output to be the same dimensionality as that of $\phi$. Given these new functions, the output of RISE is,

$$Q(s_t, a_t) = \omega((\Omega(\psi(\varphi_{\cancel{\theta}}(o_{t-k}), \varphi_{\cancel{\theta}}(o_{t-k+1}), ..., \varphi_{\cancel{\theta}}(o_t)))) \cdot \phi(o_t)), \tag{1}$$

only requiring the current observation $o_t$ to pass through $\phi$. A diagram is shown in Figure 2.

Although the idea underpinning RISE provides a cheap way to utilize recurrent models, it leaves several implementation details open for experimentation. Most notably, $\varphi_{\cancel{\theta}}$ can be any non-learnable function, rendering its choice an open-ended and task-dependent problem. One general, effective, and widely available solution is a pretrained vision model, using one of the later layers (such as the penultimate layer) as our non-learnable embedding. Throughout our evaluation and analysis, unless otherwise stated, we used the penultimate layer of a ResNet18 (He et al., 2016) image classification model. This model is fast and simple with a convenient penultimate layer size of 512; however, we do not claim this is an optimal choice, as there is a wealth of pre-trained networks to choose from.

Given that RISE uses two separate streams (LSTM stream and CNN stream), another open question is how to integrate the outputs of both streams. Various different methods to do this are explored in Section 5.2; however, we opted to upscale the LSTM stream to the same size as the CNN stream (using a linear layer), followed by a sigmoid function that then multiplies these outputs together, mimicking the idea of an attention mechanism (Chorowski et al., 2015; Luong, 2015). Lastly, for recurrent

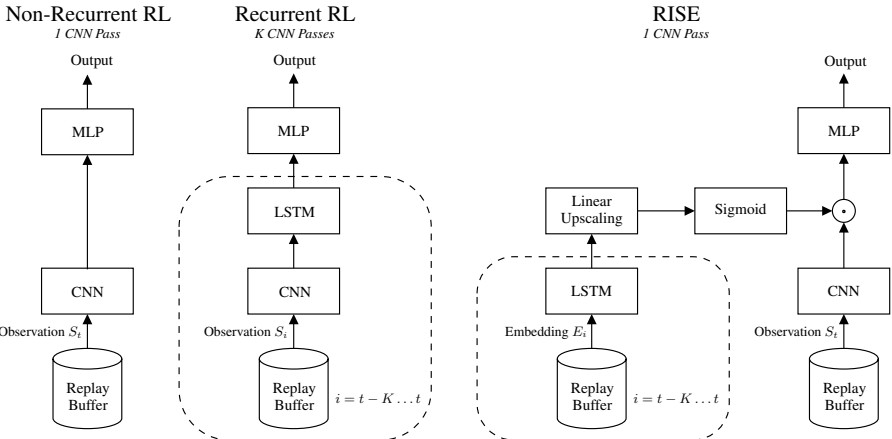

Figure 2: Diagram comparing different RL architectures, with and without a recurrent model. While we use a sigmoid function and multiplication, following the linear upscaling layer, the combination mechanism is an open design choice. Total Convolutional Neural Network (CNN) passes are the requirement to compute a single Q-value.

models, we must define how many past observations are used in training, known as the context length. Longer context lengths may perform better while slightly increasing walltime; however, the optimal context length may vary between tasks. We follow that of Kapturowski et al. (2023) using a context length of 160. Ablations for many of these decisions can be found in Section 5.

It is worth clarifying how RISE is different from the most similar prior method, R2D2 (Table 1), which would learn on trajectories rather than transitions. The primary advantage of RISE is that it requires far fewer encoder passes per batch, making it more computationally efficient in terms of speed and memory. In R2D2, the number of encoder passes is dictated by the trajectory length, burn-in length, and batch size. The disadvantage of this is that small batch sizes will cause temporally correlated batches (most samples will be from the same trajectories), small burn-in lengths will cause atypical hidden states, and small trajectory lengths will make the ratio of updates to encoder passes inefficient. Conversely, increasing these parameters will increase the number of encoder passes and, therefore computational requirements. In contrast, RISE can select any batch size and context length as needed, which is particularly useful for resource-constrained users without expensive GPUs. Like R2D2, RISE can also utilize methods to improve the hidden state, shown in Appendix H.

Table 1: Comparison of RISE and R2D2 when computing gradient steps in the off-policy setting. $\theta$ and $\theta^-$ denote the the online and target networks, $b$ is the batch size, $m$ is the sequence length for trajectory-based algorithms, $l$ is the burn-in period for R2D2, and $k$ is the context length for RISE.

| | **RISE** | **R2D2** |
|---|---|---|
| Q-values Updated per Batch | $b$ | $bm$ ($b$ sequences of length $m$) |
| Total CNN passes required | $2b$ ($b(\theta + \theta^-)$) | $2b(m + l)$ ($b(\theta + \theta^-)(m + l)$) |
| Non-Temporally Correlated Updates | $b$ ($b$ unique samples) | $b$ ($b$ unique sequences of length $m$) |
| Effect of Parameters | $b$ and $k$ can be chosen independently. | Smaller $b$ causes batches to be more temporally correlated. Reducing $l$ and $m$ will reduce the agent's context length. Lower $l$ produces more atypical hidden states. Higher $l$ reduces the ratio of encoder passes to updates. |
| Minimum Context Length For any Updated Q-Value | $k$ | $l$ (first observation in sequence only has context length $l$) |
| Original Paper Values | $b = 256, k = 160$ | $b = 64, l = 40, m = 80$ |
| -CNN Passes per Batch | 512 | 15360 |
| -Q-Values Updated per Batch | 256 | 5120 |

## 4 EVALUATION

To assess the impact of RISE on off-policy RL algorithms, we evaluate RISE when added to Beyond The Rainbow (BTR) (Clark et al., 2024) on the full Atari benchmark, as shown in Figure 1. RISE significantly improves BTR's performance, allowing it to reach performance close to state-of-the-art algorithms with significantly less walltime. To add further comparison, Figure 3 shows the BTR and Vectorized Rainbow DQN algorithms, compared to the standard recurrent and non-recurrent versions. For architecture diagrams, hyperparameters and evaluation details, see Appendices E, C and G.

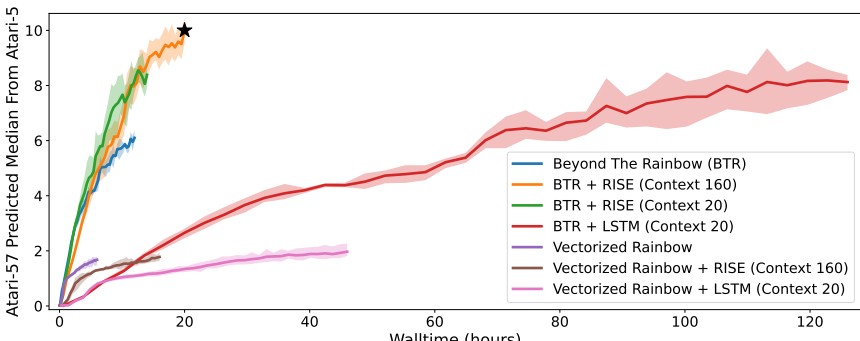

Figure 3: Performance of Vectorized Rainbow and BTR with typical LSTMS, RISE, and no recurrent model. All models were run for 200M frames. Experiments were performed on the Atari-5 subset, using a desktop PC with an RTX4090 GPU. Areas show 95% bootstrapped confidence intervals over 3 seeds. For typical LSTMs, runs were performed using a context length of 20 instead of RISE's 160 due requiring > 300 walltime hours. For compute resource details, see Appendix I.

Figure 3 shows that RISE can achieve a 22.9% performance improvement over that of a typical recurrent model and uses 84% less walltime. We found that Rainbow DQN was unable to benefit from both RISE and a typical LSTM; however, RISE still saved 40 hours of walltime in comparison. We hypothesize that as Rainbow DQN's performance is generally much lower than BTR, it is unable to benefit from recurrent models. Given the computational cost of the full Atari benchmark, results from Figure 3 and those in Section 5 use the Atari-5 subset recommended by Aitchison et al. (2023).

To be thorough in our evaluation, we tested the final BTR + RISE algorithm using four unique collections of environments: Atari-57 (Bellemare et al., 2013; Towers et al., 2024) for its extensive and well-known set of diverse and challenging tasks with history of improvement with recurrent models (Kapturowski et al., 2018), Procgen (Cobbe et al., 2020) for its procedural generation (Figure 6), Vizdoom (Kempka et al., 2016) for its 3D graphics and partial observability (Figure 7) and Miniworld (Chevalier-Boisvert et al., 2023) for a key focus on partial observability (Figure 8). Additional graphs and box plots can be found in Appendix B, while algorithm and environment hyperparameters can be found in Appendix C. We find RISE improves performance in all domains; however, improvements tend to be task-dependent. In specific environments such as *Coinrun* and *Takecover*, RISE helps dramatically, but others see no improvement, likely as they do not require long-term credit assignment or have partial observability. While RISE shows benefit, we acknowledge that other state-of-the-art algorithms in these domains exist (Cobbe et al., 2021; Hafner et al., 2023).

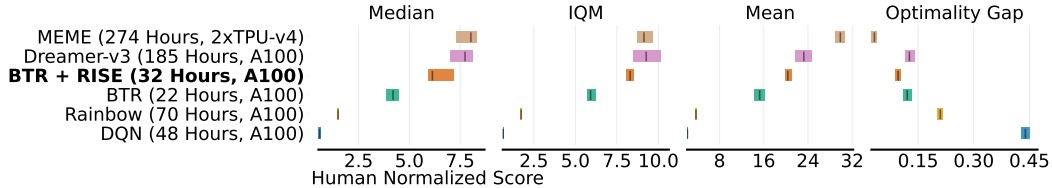

Figure 4: Box plot performance on Atari-57 of BTR+RISE (3 seeds) against other algorithms. Note scores for all algorithms use final scores at 200M, not best scores. Areas show 95% bootstrapped confidence intervals over seeds. Labels show the walltime for 200M frames, alongside the GPU used.

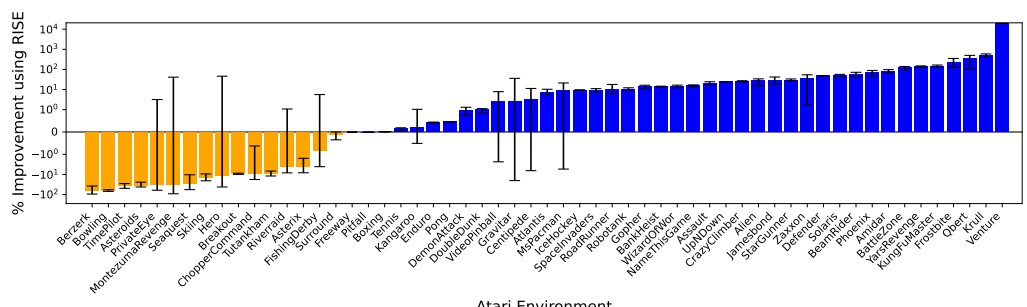

Figure 5: Percentage human normalized improvement of BTR + RISE over BTR in each of the environments in Atari-57. Error bars show 95% bootstrapped confidence intervals over 3 seeds.

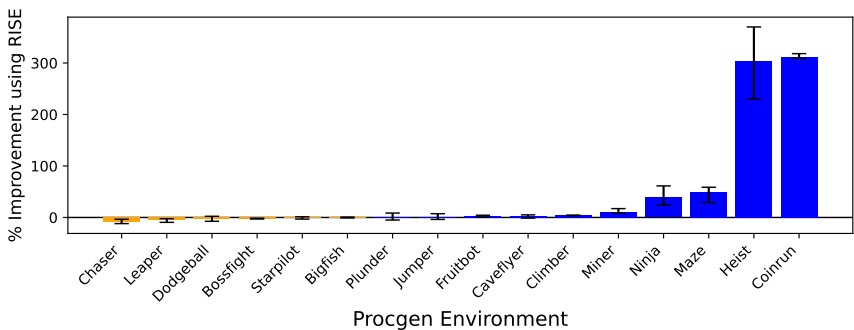

Figure 6: Percentage min-max normalized improvement of BTR + RISE over BTR in each of the environments in Procgen. Error bars show 95% bootstrapped confidence intervals over 3 seeds. While RISE had a significant improvement on some games, we found it had little impact on the total IQM, as shown in Appendix B.

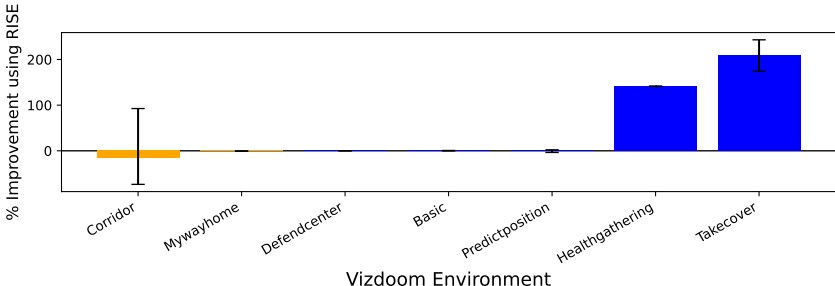

Figure 7: Performance improvement of BTR + RISE over BTR on the Vizdoom environments. Error bars show 95% bootstrapped confidence intervals over 3 seeds. For full graphs, see Appendix B.

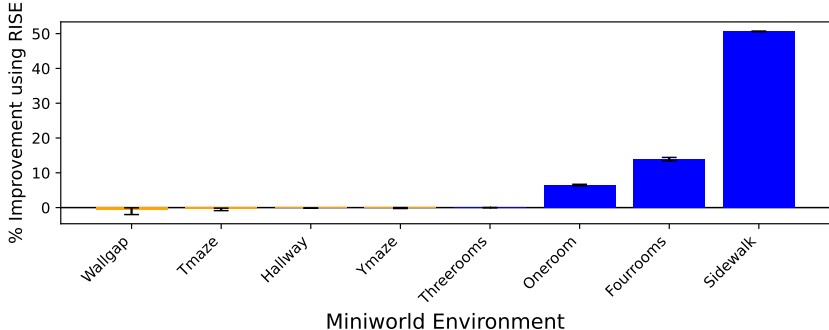

Figure 8: Performance improvement of BTR + RISE over BTR in Miniworld. Error bars show 95% bootstrapped confidence intervals over 3 seeds. For full graphs, see Appendix B.

## 5 ANALYSIS

Given RISE's performance contribution demonstrated in Section 4, we now explore the key design decisions when implementing RISE into existing algorithms. As a demonstration of how recurrent models can be used in the given environments, Appendix J provides a detailed explanation of many environments and how BTR and BTR + RISE's policies differ. Furthermore, we provide videos of BTR + RISE playing Atari-5 [1].

### 5.1 EMBEDDING METHOD ABLATIONS

A key design decision in RISE is how to implement its non-learnable encoding. This provides features for the LSTM layer, thus is important in allowing RISE to utilize a recurrent model. However, this question is open-ended; virtually any method of dimensionality reduction could be applied. The only restriction on this choice is the computational burden (for example, using large vision-language models (Radford et al., 2021; Jia et al., 2021) will slow training) and memory limitations. As RISE stores embeddings in a Replay Buffer, this limits the embedding sizes that can be used, although in practice this is rarely an issue (an embedding size of 512 and buffer size of 1M uses 2GB of memory, for any context length). There are additionally some minor GPU memory overheads from the additional model, but for the ResNet18 used in Section 4, these were negligible ($< 200$MB). As we cannot realistically test every dimensionality reduction method, we focus on a few key classes:

- Image Downscaling
- Small Pre-Trained Image Classification Models (ResNet18, as used in Section 4)
- Larger Pre-Trained Image Classification Models (EfficientNet-v2 (Tan and Le, 2021))
- Pretrained Object Detection Models (Faster RCNN (Ren et al., 2015))
- Using the Main Network's Encoder (Introduces Stale Representations)

Firstly, we test a method that does not use a pretrained encoder to verify whether pretrained models bring any value. Secondly, we test whether using larger models with larger embeddings improves performance (although the Atari benchmark may negate the benefits of this), and thirdly, whether pretrained models of different types have any significant differences. We also include two experiments using the main network's encoder. This will produce stale encodings; since the encoder is constantly changing, the encoder's output at inference would not be the same as when that encoding is sampled. To reduce the dimension of the main network's encodings, we use a fixed, untrained linear layer to downsample the encoder from 2304 (BTR's encoder output size) to 512.

Figure 9 demonstrates that RISE improves performance with all different tested encodings; however, we do find that pretrained encoders outperform naïve image downsampling. Between pretrained encoders of different sizes and initial tasks, we find no significant difference, although they may provide advantages in environments visually richer than Atari. We also want to stress that these encoders still made a substantial benefit to Atari games (and other environments), despite being trained on significantly different data (such as ImageNet). Interestingly, we also found that even when using stale encodings from the network's encoder, this only had a slightly detrimental effect on performance. Additionally, while we tested using an EMA of the main network to reduce the effect of staleness (since encodings would change much slower), we found this to be detrimental.

### 5.2 COMBINATION METHOD ABLATIONS

Given that RISE requires the use of two streams (for the learnable and non-learnable encoders), it is natural to ask how best to combine these streams. In Figures 10 and 22, we explore four different methods for this combination. We find that RISE is relatively robust to the combination mechanism; however, we found that multiplication and concatenation performed slightly better.

### 5.3 CONTEXT LENGTH

Another important hyperparameter when using a recurrent model is the context length. We review this feature for two important reasons: does performance scale with context length, and how does RISE's

---

[1]Videos at https://www.youtube.com/playlist?list=PL0BaIoEl7EKBfxRJnIXci3gVCZwftEhtb

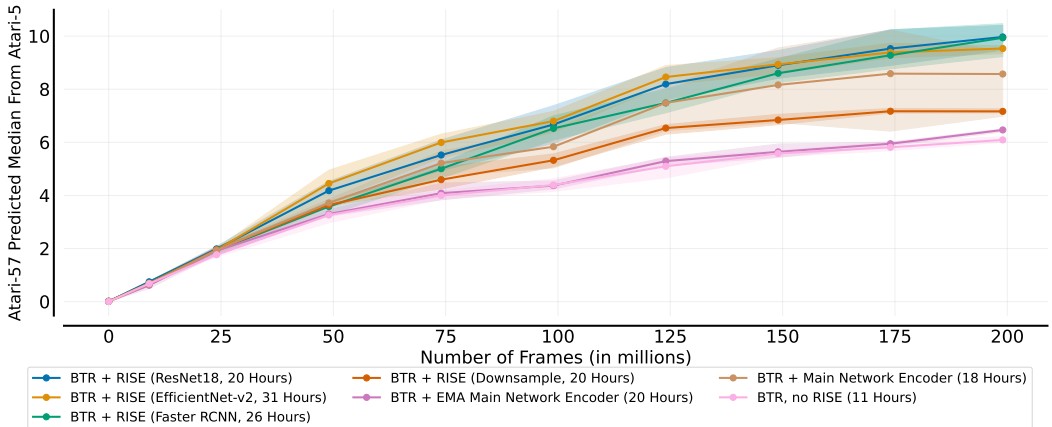

Figure 9: Performance of BTR with the RISE framework, implemented using different stored encodings. 'ResNet18' and 'EfficientNet-v2' each use the penultimate layer of their respective models, with an embedding size of 512 and 1280 respectively. 'Object Detection' uses an embedding size of 4096, randomly downprojected to 512. 'Downsample' simply downsamples and flattens the image into a 784-length encoding. The 'Main Network Encoder' uses BTR's IMPALA encoder, even though the encoder's parameters will change after the encodings are stored. EMA is an exponential moving average of this network, with smoothing factor $\alpha = 0.9997$. Additional details can be found in Appendix F. Results are averaged over three seeds on the Atari-5 benchmark, with shaded areas indicating 95% confidence intervals. Legend shows walltime on a desktop PC with an RTX4090.

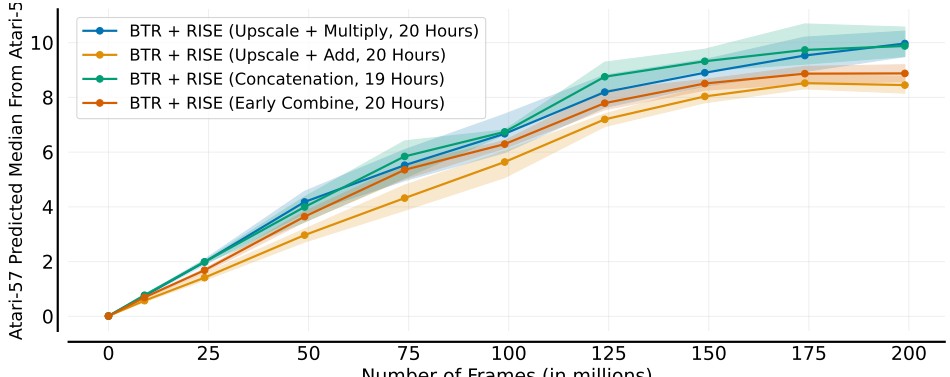

Figure 10: Analysis of BTR with our proposed RISE framework, with varying mechanisms to combine the two streams on the Atari-5 benchmark. Upscale + Multiply is the variant used throughout the paper, which projects the LSTM's output to the same size as the learnable encoder's output (via a linear layer), passes this through a sigmoid function, and then multiplies the two streams. Upscale + Add is the same, but adds the streams. Concatenation simply concatenates the output of the LSTM and learnable encoder. Lastly, Early Combine is the same as the multiply variant, but additionally downsamples the learnable encoder's output to the encoding's size, multiplies them together and passes through a sigmoid **before** passing through the LSTM . All runs used 3 seeds, with shaded areas showing 95% confidence intervals. Legend shows walltime on a desktop PC with an RTX4090.

walltime scale with different context lengths? Figure 11 shows that RISE can achieve increased performance even with small context lengths such as 20, but can continue to gain performance until around 80-160 (results are similar to Badia et al. (2020), likely due to the task). While RISE suffers from linearly increasing walltime as context length increases, it can still feasibly use long context lengths such as 160. Throughout this paper, unless otherwise specified, we use a context length of 160. An additional ablation on LSTM size can be found in Appendix D.

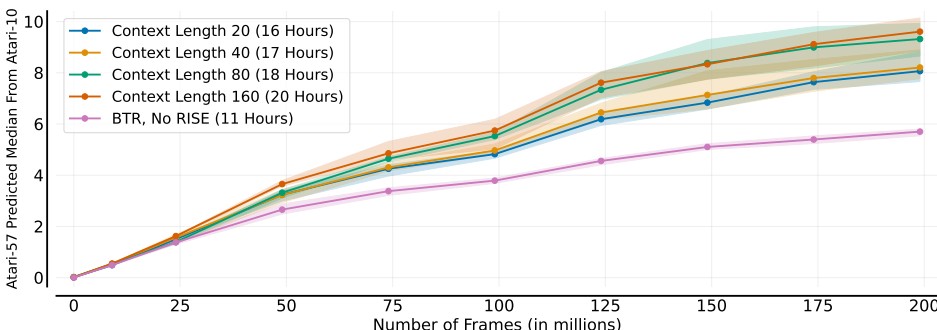

Figure 11: Analysis of BTR with our proposed RISE framework, with varying context lengths on the Atari-10 benchmark. All runs used 3 seeds, with shaded areas showing 95% confidence intervals. Legend shows walltime on a desktop PC with an RTX4090.

## 5.4 Sequence Model Architecture

In RL, LSTMs have been the most popular architecture for overcoming partial observability. Nevertheless, many alternatives exist. We explore the performance of RISE implemented using different techniques, such as Transformers (Vaswani et al., 2017; Parisotto et al., 2020) which have seen widespread use in other areas of Deep Learning, Fast and Forgetful memory (FFM) (Morad et al., 2023) which was specifically designed for RL, and MAMBA (Gu and Dao, 2023) which enjoys linear scaling with context length. Figure 12 shows that LSTMs still retain the best performance and fastest walltime. We found transformers and MAMBA to perform marginally worse, but increased walltime[2]. FFM gave the lowest performance, but matched the walltime of LSTMs. While we found LSTMs to give the best results, additional tuning for other architectures may yield better results. These models show a stark contrast between RL and other areas of Deep Learning, particularly since RL typically uses significantly shorter context lengths (160 instead of greater than 32,000 (Team et al., 2023)). Furthermore, we also tested naïvely using an LSTM while only using a pre-trained encoder, rather than using two separate streams, with no additional changes. We found that this performed significantly worse than using two streams; however, this may be possible with future work.

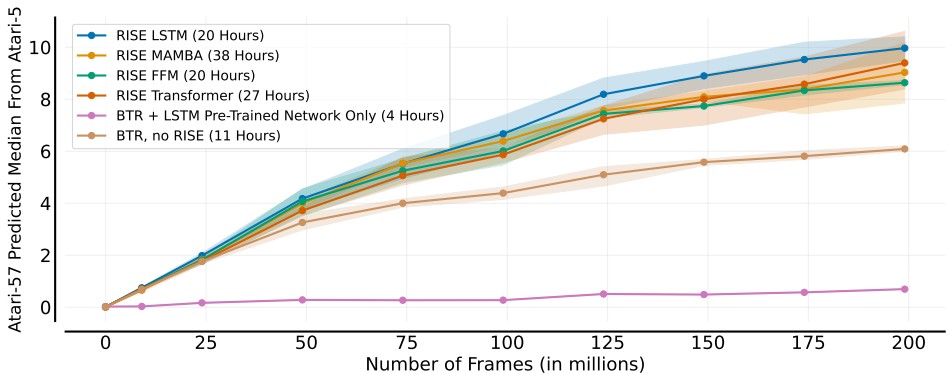

Figure 12: Analysis of BTR with our proposed RISE framework, using different architectures to handle sequences of inputs on the Atari-5 benchmark. All runs used 3 seeds, with shaded areas showing 95% confidence intervals. Legend shows walltime on a desktop PC with an RTX4090.

## 5.5 R2D2 Comparison

As described in Section 3, the most similar prior work to ours is R2D2 (Kapturowski et al., 2018), which learned from trajectories. To compare the performance of these algorithms, we test BTR + R2D2 with different sequence lengths ($m$) and and batch sizes ($b$) in Figures 13 and 25. In R2D2, the

---

[2]It is worth noting we use MAMBA 1, not the more modern MAMBA 2 (Dao and Gu, 2024)

total number of state updates (and CNN encoder passes) is equal to $bm$, since every state from the sequence is updated. We find that naïvely applying R2D2 to BTR reduced performance, but improves as the batch size increases. This supports the idea that temporally correlated states (which RISE helps to avoid) are detrimental to performance.

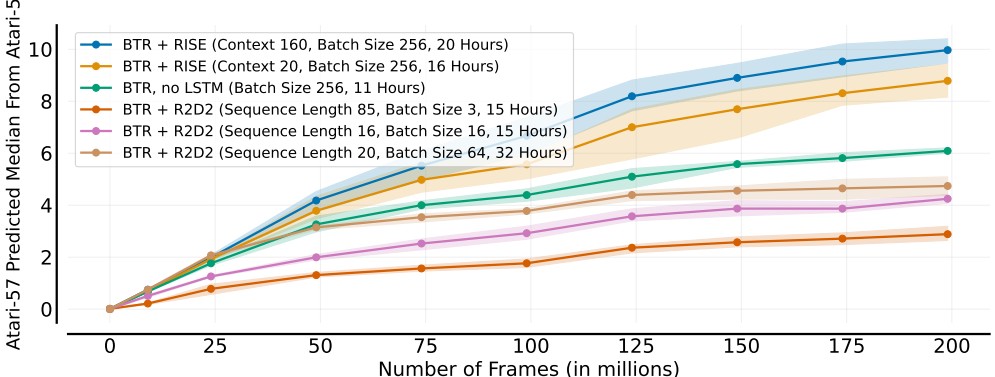

Figure 13: BTR with RISE and R2D2, on the Atari-5 benchmark. All runs used 3 seeds, with shaded areas showing 95% confidence intervals, and walltime based on a desktop PC with an RTX4090.

# 6 RELATED WORK

RL has a long history with both recurrent models (Hausknecht and Stone, 2015; Morad et al., 2024; Lu et al., 2023) and off-policy algorithms (Mnih, 2013; Munos et al., 2016; Haarnoja et al., 2018). While their combination is not novel (Kapturowski et al., 2018; Badia et al., 2020; Kapturowski et al., 2023; Luo et al., 2024), there is little prior work on doing so in a computationally efficient way for the off-policy setting with expensive encoders. Previous computationally efficient RL has typically avoided recurrent models (Schmidt and Schmied, 2021; Clark et al., 2024), or opted to use on-policy algorithms (Schulman et al., 2017; Mnih et al., 2016) which don't face the same issues. While there has been prior work to improve computational efficiency via stored embeddings (Chen et al., 2021) and using pretrained models (Chen et al., 2024), this was unrelated to recurrent models and did not look to leverage the advantages of both learnable and non-learnable embeddings simultaneously. There have also been previous attempts at using embeddings trained on task-specific data (Schwarzer et al., 2021; Kim et al., 2024); however, these have the obvious downsides of requiring this data before training and being unable to adapt to environments that change during training.

# 7 CONCLUSION AND FUTURE WORK

We present a new framework, RISE, that enables the compute-efficient and high-performance use of recurrent models in any image-based off-policy RL algorithm. Prior algorithms have either been forced to bear a heavy computational burden or sacrifice the advantages of using recurrent models. Our experiments demonstrated that RISE provides a significant performance boost on a wide variety of tasks, while only adding a fraction of the compute cost compared to prior work incorporating recurrent models in image-based off-policy RL.

We acknowledge that the method presented in this paper is specifically designed for image-based tasks, limiting its breadth of application; that said, RISE is not limited to only images, but rather any task with expensive early encoder layers. For example, RISE could be used with transformers for text-based tasks, or CNNs for audio-based tasks. Lastly, RISE naturally opens up the door for future work regarding the implementation of its non-learnable encoding. We also acknowledge that the use of a fixed, pre-trained encoder may limit the model's ability to extract useful features in some environments. In this sense, RISE acts as a trade-off, allowing greater compute-efficiency in exchange for less adaptability if the encoder does not capture the relevant features. While we take the naïve approach of using a simple pre-trained encoder, work combining RISE with existing research on pre-trained representations may hold further benefit. In addition to allowing recurrent models, RISE may also improve performance by using the knowledge from a pre-trained encoder.

## 8 ETHICS STATEMENT

Given that recurrent off-policy RL yields some of the highest-performing agents, this work has the potential to increase accessibility for labs and practitioners who lack access to high-performance compute facilities or those who wish to use them sparingly. Increasing accessibility does carry the risk of misuse by bad actors; however, we believe this risk is justified by the potential positive impact.

## 9 REPRODUCIBILITY STATEMENT

We have made significant efforts to make our work reproducible. Most importantly, we have made our code open source with instructions on how to run our work. Furthermore, we included detailed diagrams (Figures 2, 31) and tables (Tables 3, 7) on the architectures and hyperparameters used in our work, such that anyone can implement our proposed technique.

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

## A  FULL RESULTS TABLE

Table 2: Scores at 200M Frames on the Atari-57 benchmark. These use the final score, not the best score as done in some previous work. All scores here use sticky actions.

| Game | Random | Human | DQN | Rainbow | BTR | BTR + RISE | Dreamer-v3 | MEME |
|---|---|---|---|---|---|---|---|---|
| Alien | 228 | 7128 | 1938 | 3262 | 15807 | 21485 | 15089 | **31589** |
| Amidar | 6 | 1720 | 1117 | 2530 | 6727 | **16228** | 4444 | 7943 |
| Assault | 222 | 742 | 1528 | 3492 | 12565 | 19310 | 29991 | **34973** |
| Asterix | 210 | 8503 | 2990 | 15513 | 240929 | 323003 | 264235 | **846371** |
| Asteroids | 719 | 47389 | 846 | 1449 | 112398 | 72149 | **401218** | 208806 |
| Atlantis | 12850 | 29028 | 863260 | 749800 | 810604 | 864729 | 1438294 | **1475324** |
| BankHeist | 14 | 753 | 573 | 1104 | 1583 | 1622 | 1037 | **4333** |
| BattleZone | 2360 | 37188 | 18219 | 36337 | 152690 | 307730 | 615467 | **769218** |
| BeamRider | 364 | 16926 | 5596 | 6619 | 65300 | **131462** | 35597 | 49517 |
| Berzerk | 124 | 2630 | 468 | 858 | 5044 | 919 | 8101 | **21491** |
| Bowling | 23 | 161 | 26 | 44 | 30 | 33 | 247 | **264** |
| Boxing | 0 | 12 | 77 | 97 | **100** | **100** | **100** | **100** |
| Breakout | 2 | 30 | 96 | 119 | 602 | 443 | 439 | **605** |
| Centipede | 2091 | 12017 | 2335 | 6636 | 43811 | 69042 | **704016** | 60273 |
| ChopperCommand | 811 | 7388 | 2480 | 13240 | 786104 | **892682** | 859671 | 22759 |
| CrazyClimber | 10780 | 35829 | 107351 | 150181 | 133675 | 168990 | 190475 | **281072** |
| Defender | 2874 | 18689 | 6386 | 59845 | 345600 | 489402 | **588970** | 540128 |
| DemonAttack | 152 | 1971 | 2795 | 18003 | 135114 | 135875 | **143473** | 132572 |
| DoubleDunk | -19 | -16 | -5 | 22 | 23 | 23 | **24** | 23 |
| Enduro | 0 | 860 | 649 | 2124 | 2343 | 2360 | **2367** | 2338 |
| FishingDerby | -92 | -39 | 1 | 42 | 55 | 53 | **84** | 64 |
| Freeway | 0 | 30 | 0 | **34** | **34** | **34** | **34** | **34** |
| Frostbite | 65 | 4335 | 909 | 8004 | 16619 | 28521 | 1914 | **119625** |
| Gopher | 258 | 2412 | 5566 | 11294 | 50253 | 65761 | 94908 | **99451** |
| Gravitar | 173 | 3351 | 195 | 1229 | 4304 | 3463 | **13471** | 12903 |
| Hero | 1027 | 30826 | 13621 | 43297 | 21486 | 38348 | **45286** | 30262 |
| IceHockey | -11 | 1 | -6 | -1 | 37 | 49 | **68** | 36 |
| Jamesbond | 29 | 303 | 578 | 805 | 14919 | 45252 | 26926 | **130457** |
| Kangaroo | 52 | 3035 | 10668 | 13810 | 13792 | 12918 | 11197 | **14842** |
| Krull | 1598 | 2666 | 6027 | 4771 | 11045 | 45307 | 94674 | **137007** |
| KungFuMaster | 258 | 22736 | 22388 | 26463 | 44509 | 125545 | 158993 | **175455** |
| MontezumaRevenge | 0 | 4753 | 0 | 0 | 0 | 0 | 400 | **2808** |
| MsPacman | 307 | 6952 | 3609 | 4167 | 8953 | 12132 | 23381 | **27644** |
| NameThisGame | 2292 | 8049 | 7083 | 8931 | 24778 | 32300 | **93675** | 27392 |
| Phoenix | 761 | 7243 | 5048 | 10484 | 169692 | 435229 | **463389** | 445134 |
| Pitfall | -229 | 6464 | -38 | -9 | 0 | 0 | 0 | **619** |
| Pong | -21 | 15 | 19 | 20 | **21** | **21** | 16 | 19 |
| PrivateEye | 25 | 69571 | 230 | 5666 | 100 | 100 | 5374 | **69549** |
| Qbert | 164 | 13455 | 10127 | 16143 | 44818 | 70136 | **110343** | 57401 |
| Riverraid | 1338 | 17118 | 11660 | 21517 | 22382 | 22410 | **58582** | 36671 |
| RoadRunner | 12 | 7845 | 39587 | 53526 | 222744 | 373174 | 129091 | **566126** |
| Robotank | 2 | 12 | 58 | 65 | 77 | 87 | **130** | 97 |
| Seaquest | 68 | 42055 | 2091 | 6028 | 409991 | 290096 | 7621 | **562431** |
| Skiing | -17098 | -4337 | -16695 | -30133 | -9557 | -10169 | -29535 | **0** |
| Solaris | 1236 | 12327 | 1556 | 2042 | 0 | 5801 | 6042 | **12103** |
| SpaceInvaders | 148 | 1669 | 1803 | 2479 | 41553 | **47308** | 14471 | 26123 |
| StarGunner | 664 | 10250 | 49240 | 56502 | 478095 | **685953** | 288733 | 179035 |
| Surround | -10 | 6 | -7 | **10** | **10** | **10** | 8 | **10** |
| Tennis | -24 | -8 | -5 | -1 | **24** | 23 | 0 | 23 |
| TimePilot | 3568 | 5229 | 3696 | 11577 | 103499 | 66707 | **177400** | 126297 |
| Tutankham | 11 | 168 | 118 | 241 | 315 | 266 | **355** | 334 |
| UpNDown | 533 | 11693 | 7031 | 21187 | 397876 | 453279 | **591438** | 449387 |
| Venture | 0 | 1188 | 55 | 1475 | 0 | 1789 | 0 | **2301** |
| VideoPinball | 0 | 17668 | 75058 | 468055 | 589066 | 495639 | **997074** | 777329 |
| WizardOfWor | 564 | 4756 | 369 | 7214 | 42570 | 54280 | **105600** | 54976 |
| YarsRevenge | 3093 | 54577 | 24556 | 50529 | 177431 | 451571 | **907564** | 428038 |
| Zaxxon | 32 | 9173 | 3980 | 14466 | 40525 | 73160 | 62964 | **97816** |
| Median (↑) | 0.000 | 1.000 | 0.588 | 1.493 | 4.199 | 6.134 | 7.74 | **8.029** |
| IQM (↑) | 0.000 | 1.000 | 0.657 | 1.736 | 5.918 | 8.299 | **9.268** | 9.131 |
| Mean (↑) | 0.000 | 1.000 | 2.195 | 3.737 | 15.264 | 20.359 | 23.257 | **29.861** |
| Optimality Gap (↓) | 0.000 | 1.000 | 0.439 | 0.21 | 0.121 | 0.097 | 0.127 | **0.032** |
| >Human | - | - | 19 | 40 | 49 | 50 | 49 | **52** |

# B FULL RESULTS GRAPHS

## B.1 ATARI

Many of the figures from the main paper used either the entire Atari suite, or used the Atari-5 or Atari-10 subsets. In this subsection, we present the graphs for individual games, or using a different X-axis. Figures 14 and 15 show BTR + RISE on individual games from the full Atari suite. Figure 17 shows the same data as Figure 1 but using frames for the X-axis rather than walltime. Figure 16 shows BTR's performance on the 26 game Atari subset, commonly used for benchmarking sample-efficient algorithms. We include this figure to show that while sample efficient algorithms use fewer frames, their walltime-to-performance does not compare against BTR + RISE (nor do they attempt to). Figures 18 and 19 show the IQM and indivdual game performance of BTR and vectorized Rainbow by themselves, with RISE, and using a typical LSTM. Figures 20, 21, 23 and 24 show individual game performance for each of the ablations run in our analysis.

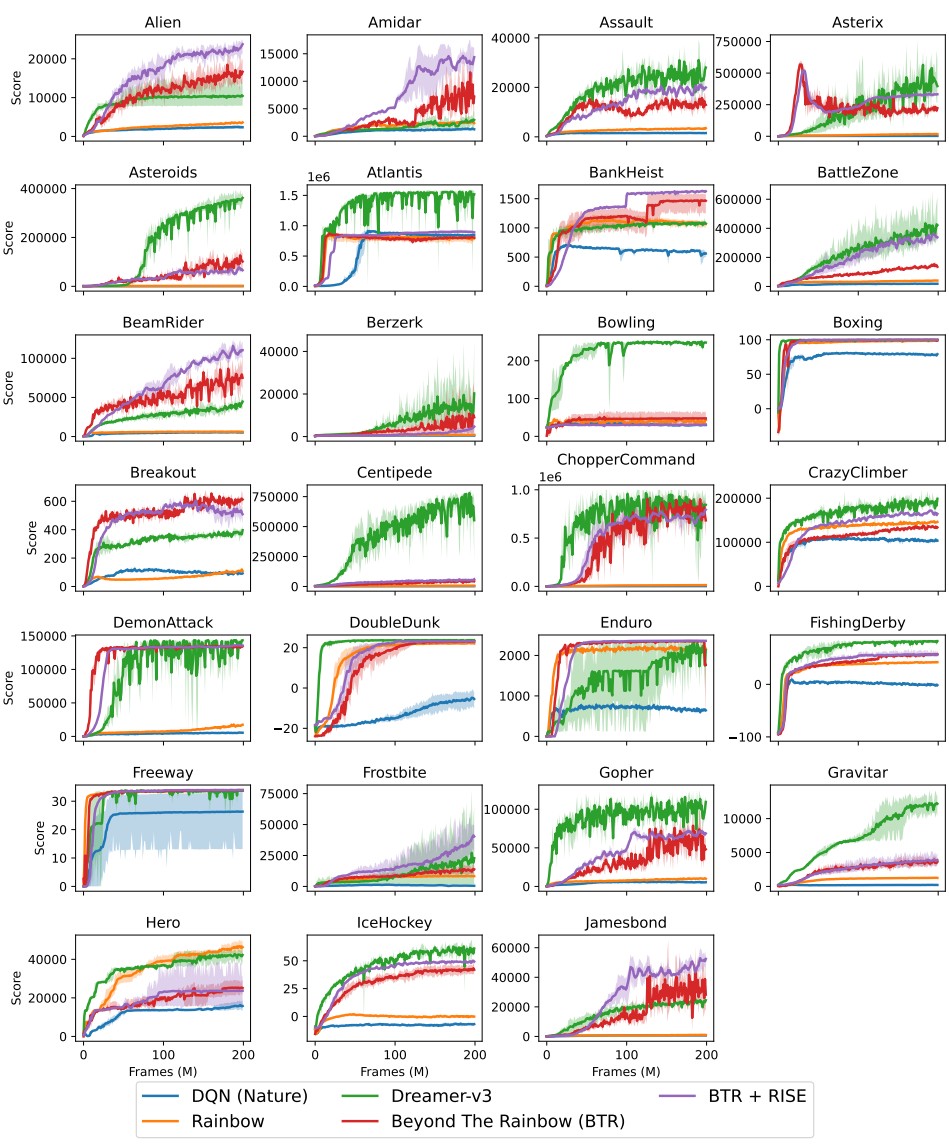

Figure 14: Graph showing the first half of games in the Atari-55 benchmark. We did not have full curve data for *Defender* and *Surround* for some algorithms, hence we could not include all 57 games. Shaded areas show 95% confidence intervals, using 3 seeds.

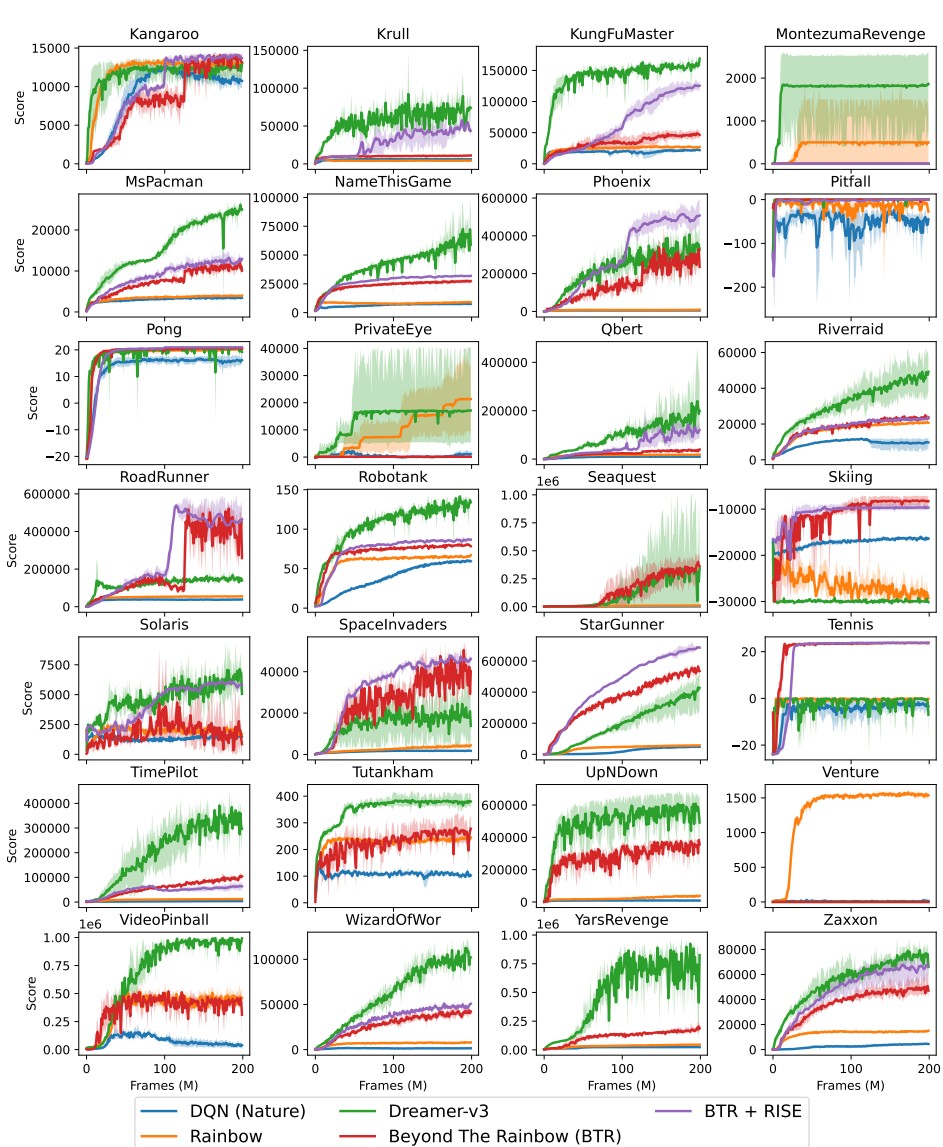

Figure 15: Graph showing the second half of games in the Atari-55 benchmark. We did not have full curve data for *Defender* and *Surround* for some algorithms, hence we could not include all 57 games. Shaded areas show 95% confidence intervals, using 3 seeds.

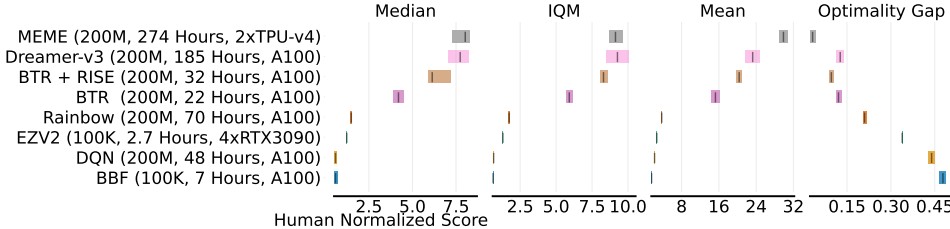

Figure 16: Graph showing the 26 Atari environments from the sample-efficient benchmark. In brackets, we show the number of frames used, the walltime to complete that number of frames, and the hardware used. We find that sample-efficient algorithms do not have good walltime efficiency (this is not the objective of such algorithms). EZV2 refers to EfficientZero-v2 (Wang et al., 2024) and BBF refers to Bigger, Better, Faster (Schwarzer et al., 2023).

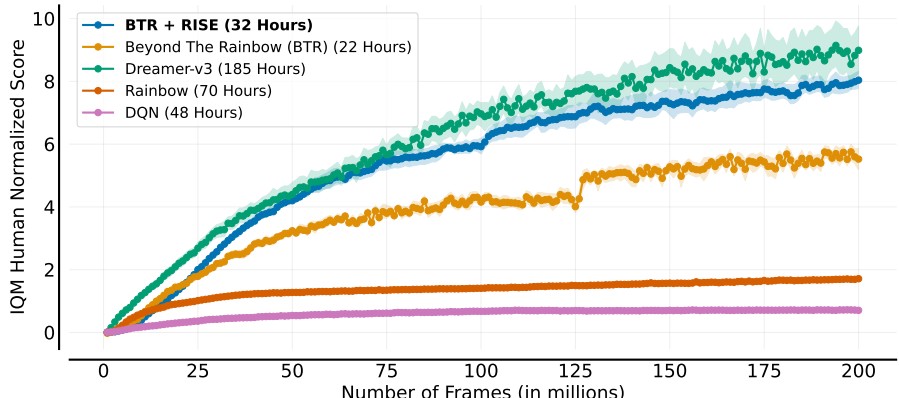

Figure 17: Graph showing performance of different algorithms on the Atari-55 benchmark similarly to Figure 1, but uses number of frames instead of A100 walltime hours as the X-axis.

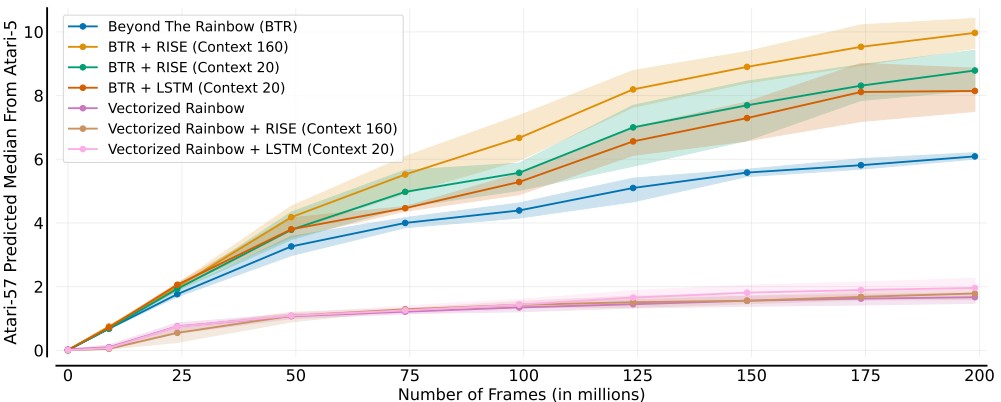

Figure 18: Graph showing the Atari-5 scores of BTR and Vectorized Rainbow with no additions, RISE, or an LSTM (see Figure 3). Shaded areas show 95% confidence intervals, using 3 seeds.

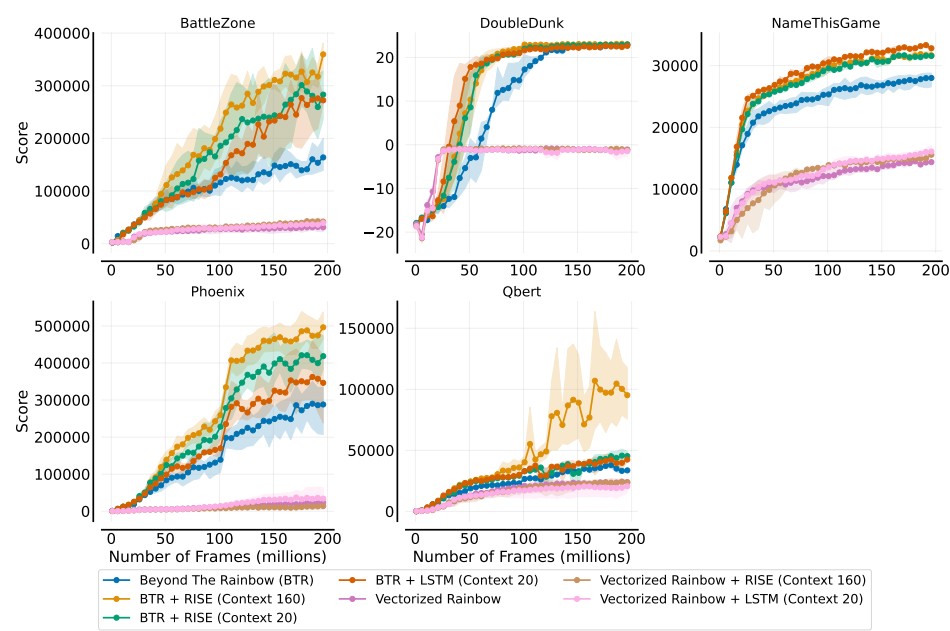

Figure 19: Graph showing the individual game of BTR and Vectorized Rainbow with no additions, RISE, or an LSTM on Atari-5. Shaded areas show 95% confidence intervals, using 3 seeds.

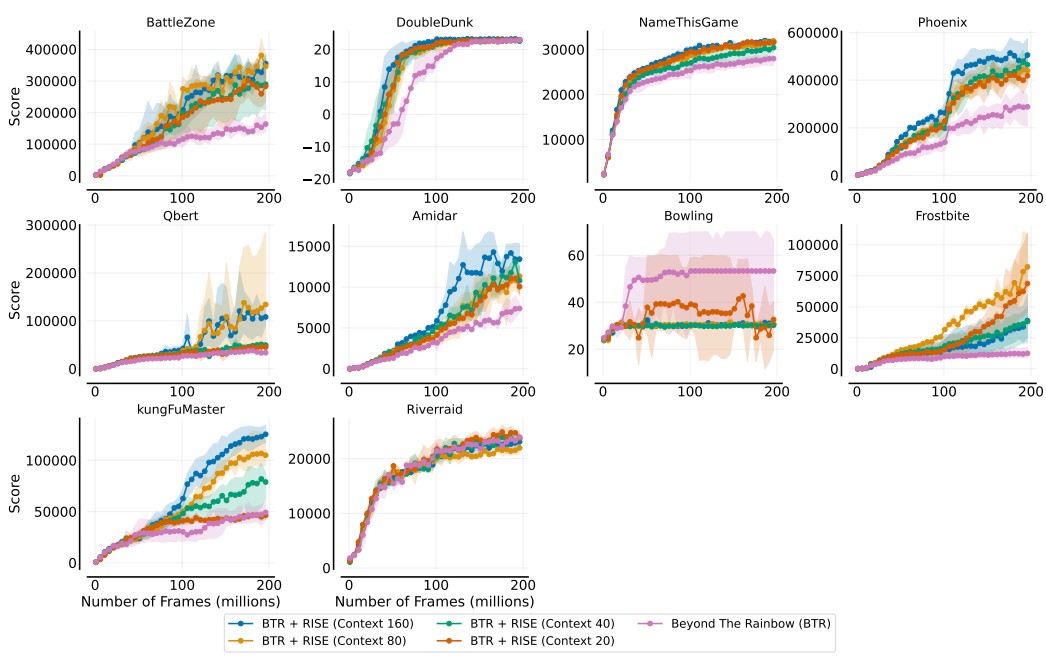

Figure 20: Graph showing the individual game scores of BTR and BTR + RISE with different context lengths on the Atari-10 benchmark (see Figure 11). Shaded areas show 95% confidence intervals, using 3 seeds.

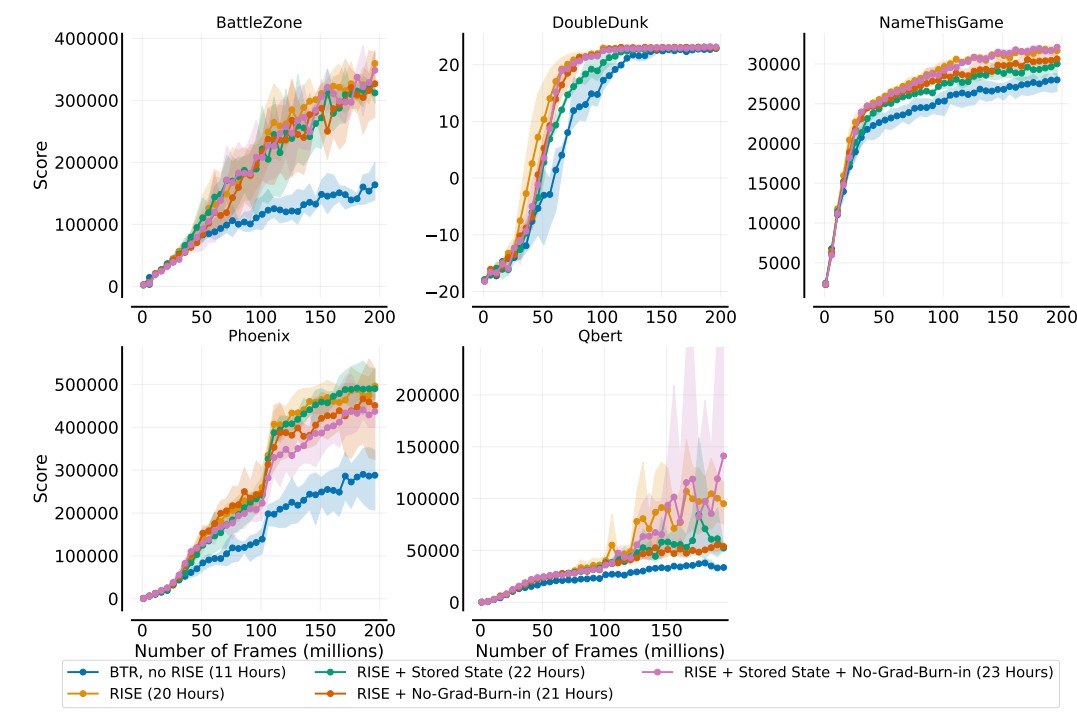

Figure 21: Graph showing the individual game scores of BTR and BTR + RISE with different off-policy recurrent paradigms on the Atari-5 benchmark (see Figure 36). Shaded areas show 95% confidence intervals, using 3 seeds.

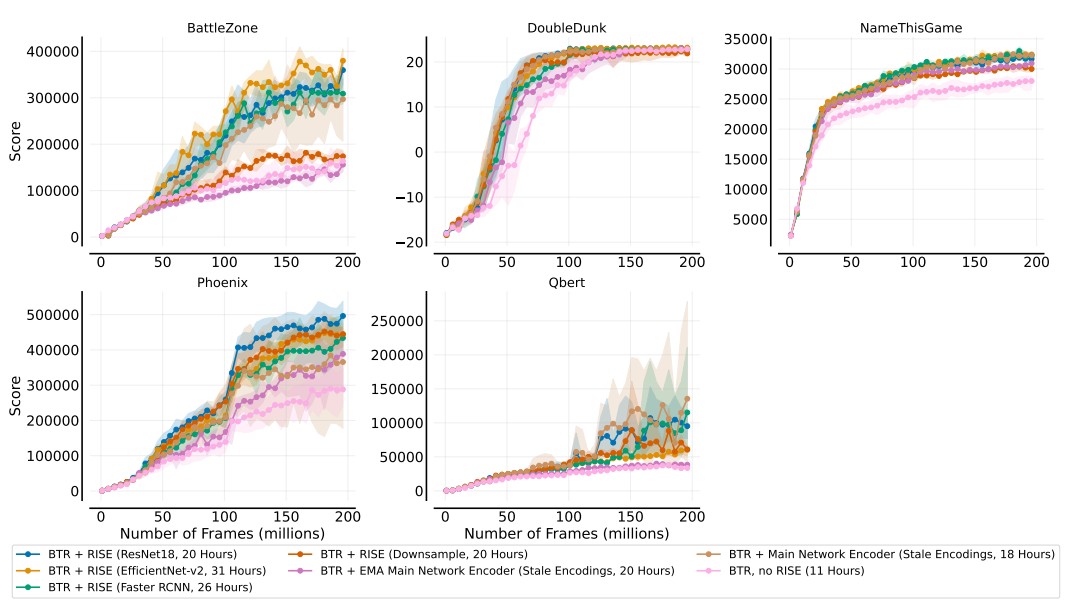

Figure 23: Graph showing the individual game scores of BTR and BTR + RISE with encoders for providing RISE's embeddings on the Atari-5 benchmark (see Figure 9). Shaded areas show 95% confidence intervals, using 3 seeds.

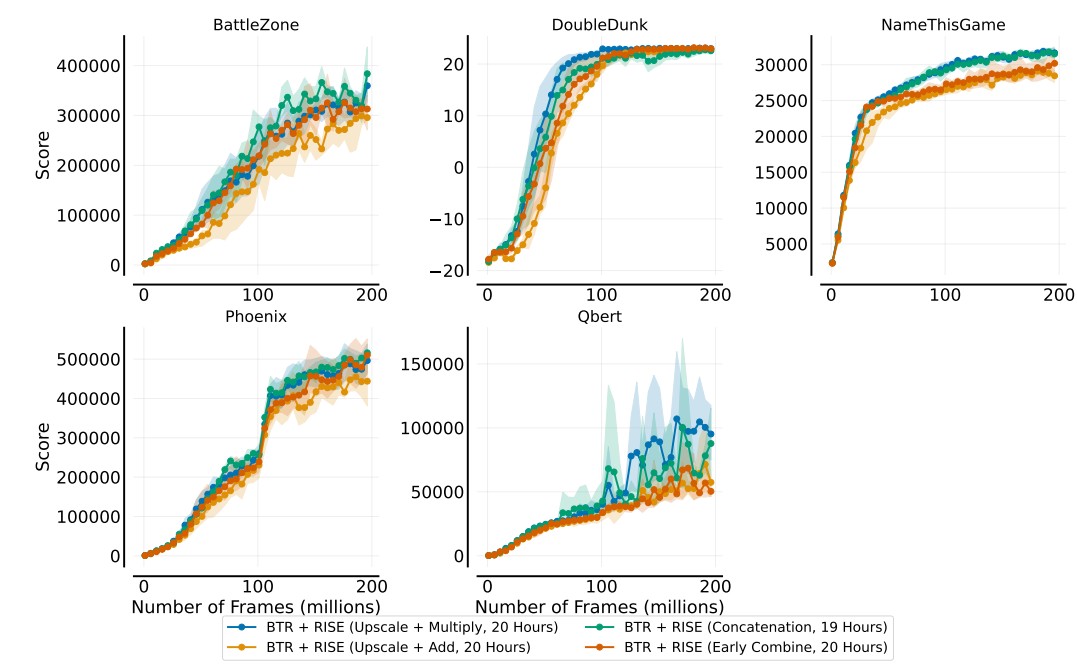

Figure 22: Graph showing the individual game scores of BTR and BTR + RISE with combination mechanisms on the Atari-5 benchmark (see Figure 10). Shaded areas show 95% confidence intervals, using 3 seeds.

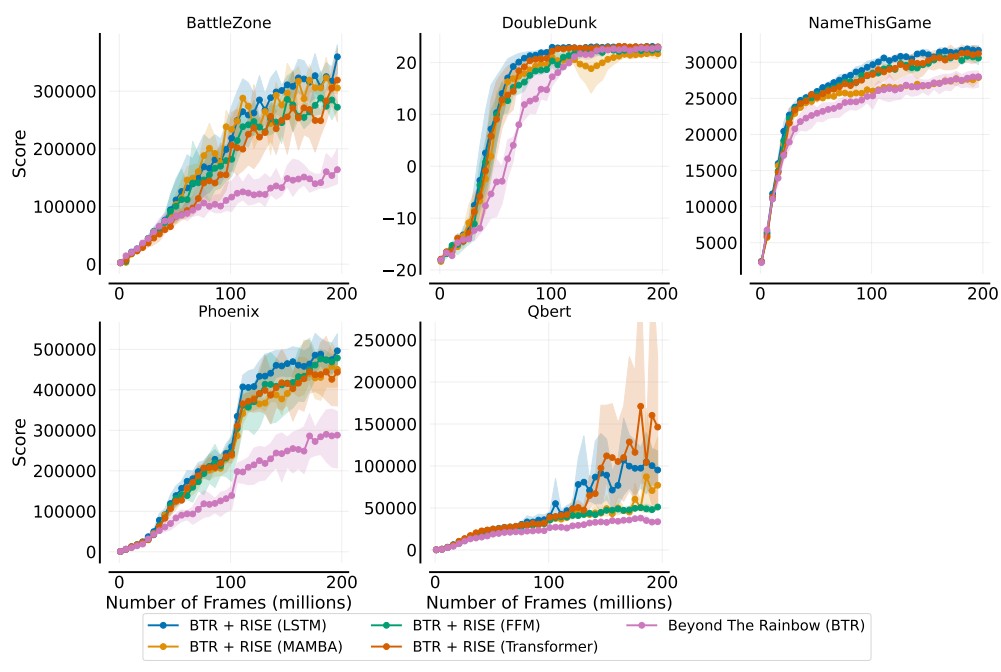

Figure 24: Graph showing the individual game scores of BTR and BTR + RISE with different architectures for handling sequences of input on the Atari-5 benchmark (see Figure 12). Shaded areas show 95% confidence intervals, using 3 seeds.

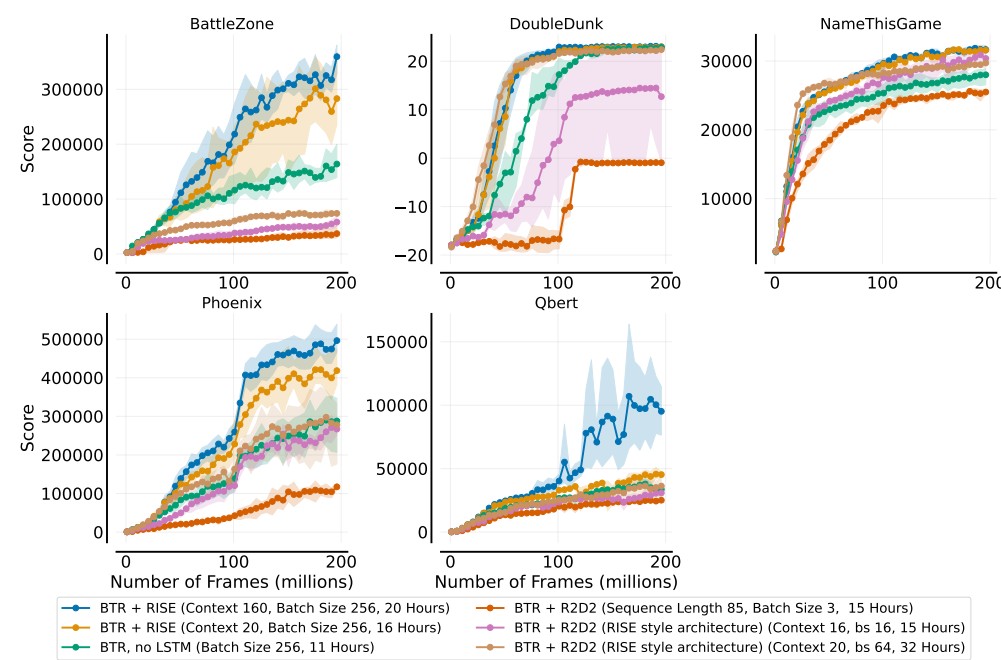

Figure 25: Graph showing the individual game scores of RISE and R2D2 with different with different parameters on the Atari-5 benchmark (see Figure 12). Shaded areas show 95% confidence intervals, using 3 seeds.

## B.2 PROCGEN

When evaluating on Procgen, we found that BTR + RISE only made an impact on a few environments, and generally left others with little difference. This may be due to the nature of some environments not being able to benefit from recurrent models. While Figure 6 shows the individual change between games, Figure 26 shows the total IQM performance. The IQMs are very similar as RISE only helped a some specific environments (such as *Coinrun* and *Heist*). For individual curves for each game, see Figure 27.

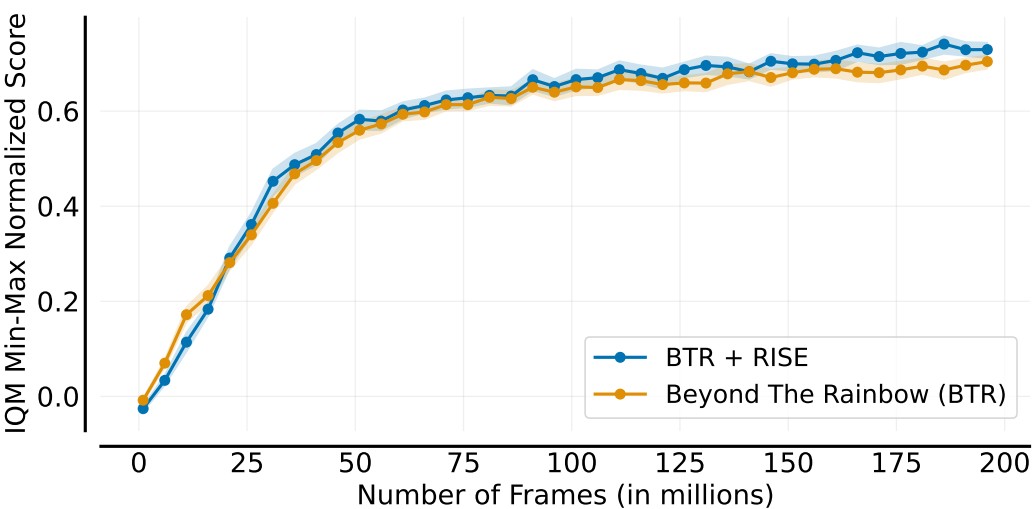

Figure 26: Graph showing the IQM min-max normalized scores of BTR and BTR + RISE on all 16 Procgen environments, using the hard distribution. Shaded areas show 95% confidence intervals, using 3 seeds.

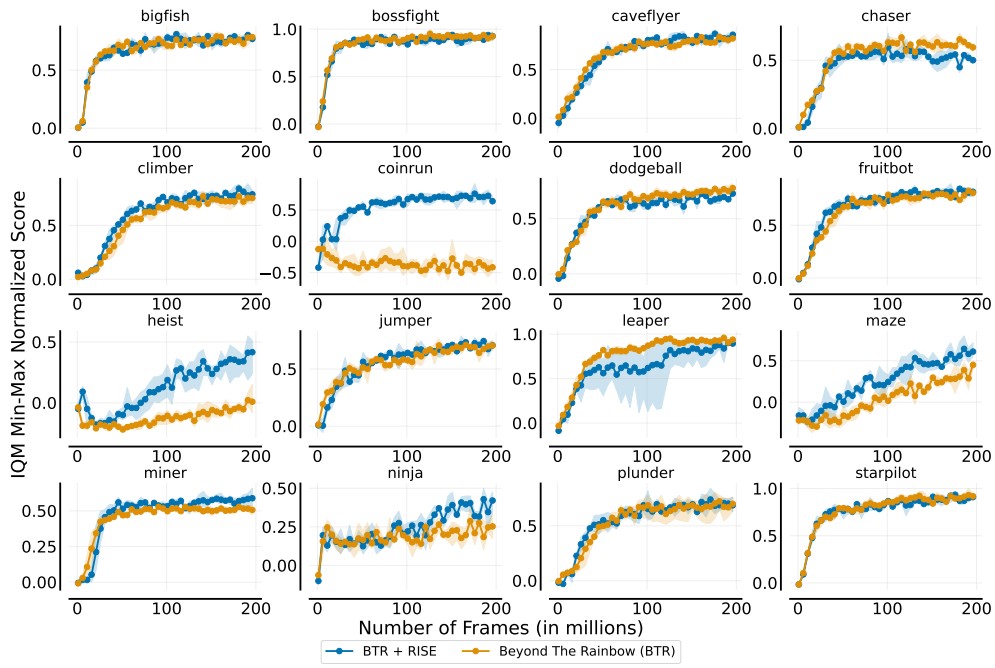

Figure 27: Graph showing the individual performance for each game on the Procgen benchmark for both BTR and BTR + RISE. Shaded areas show 95% confidence intervals, using 3 seeds.

### B.3 VIZDOOM

When evaluating on Vizdoom (Figure 28), we again found performance to be very environment specific. In some environments such as *HealthGathering* and *TakeCover*, BTR + RISE drastically outperformed BTR. However, on some other environments BTR + RISE exhibited some instability, causing performance to fluctuate during training.

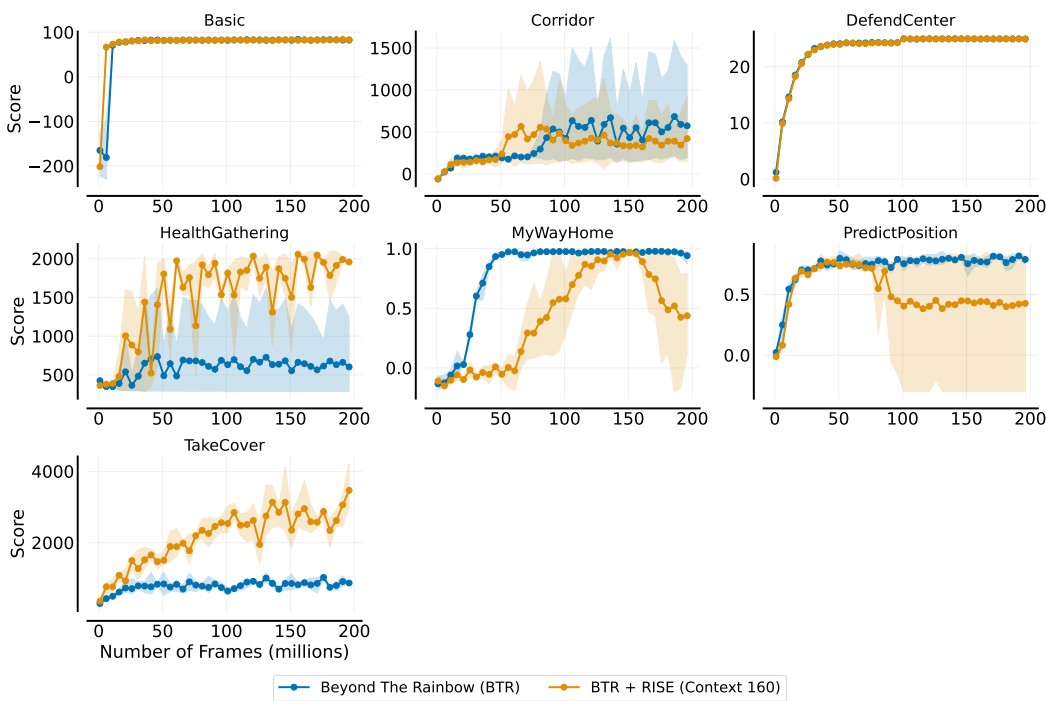

Figure 28: Graph showing the individual performance for each game on the Vizdoom benchmark for both BTR and BTR + RISE. Shaded areas show 95% confidence intervals, using 3 seeds.

### B.4 MINIWORLD

When evaluating on Miniworld (Figure 29), we found results to be similar to those of VizDoom, with specific environment improvements. *FourRooms* showed a consistent significance improvement, *OneRoom* showed a consistent minor improvement, and *Sidewalk* showed a large improvement; however, BTR had very high uncertainty in this environment. The remaining environments were solved almost 100% of the time by both BTR and BTR + RISE, except *ThreeRooms*, which failed due to lack of exploration. Again, RISE did exhibit some instability in some environments during training, but it eventually settled.

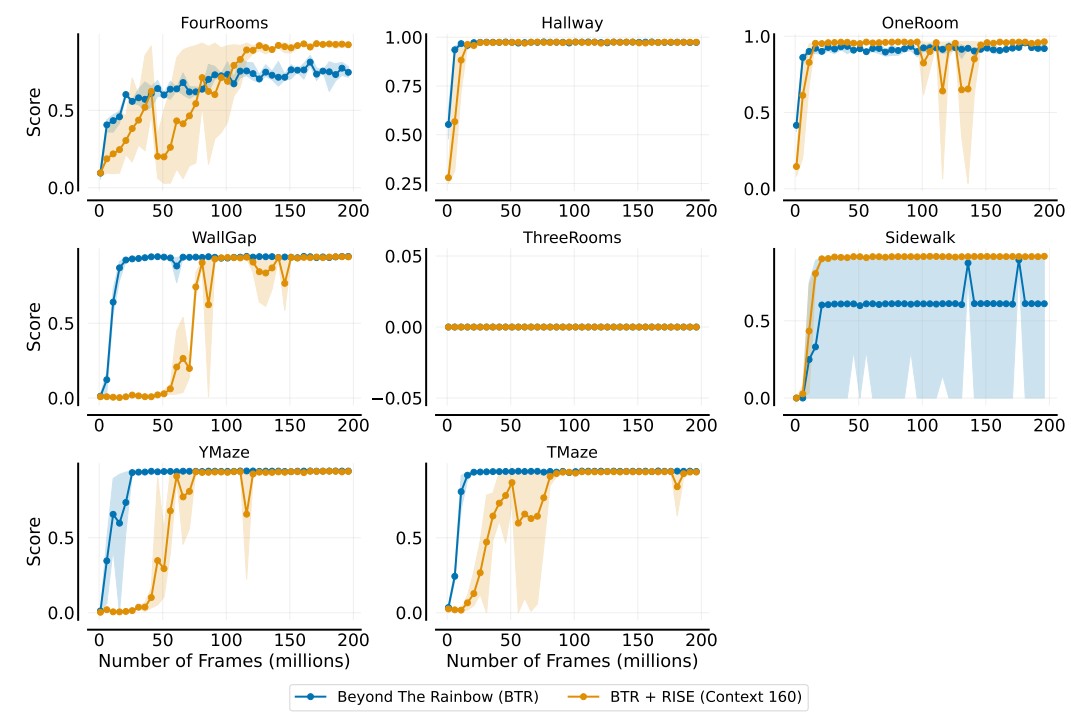

Figure 29: Graph showing the individual performance for each game on the Miniworld benchmark for both BTR and BTR + RISE. Shaded areas show 95% confidence intervals, using 3 seeds.

# C  HYPERPARAMETERS

## C.1  ENVIRONMENT HYPERPARAMETERS

Table 3: Environment Details for Atari Experiments.

| Hyperparameter | Value |
| --- | --- |
| Grey-Scaling | True |
| Observation down-sampling | 84x84 |
| Frames Stacked | 4 |
| Reward Clipping | [-1, 1] |
| Terminal on loss of life | False |
| Life Information | False |
| Max frames per episode | 108K |
| Sticky Actions | True |
| Version | ALE Version 5 |
| Vectorization Method | Gymnasium Async |

Table 4: Environment Details for Procgen Experiments.

| Hyperparameter | Value |
| --- | --- |
| Grey-Scaling | False |
| Observation Size | 64x64 |
| Frames Stacked | 1 |
| Reward Clipping | False |
| Max frames per episode | 108K |
| Distribution Mode | Hard |
| Number of Unique Levels (Train & Test) | Unlimited |

Table 5: Environment Details for Vizdoom Experiments.

| Hyperparameter | Value |
| --- | --- |
| Grey-Scaling | True |
| Observation Size | 84x84 |
| Frames Stacked | 4 |
| Reward Clipping | True |
| Max frames per episode | 108K |

Table 6: Environment Details for Miniworld Experiments.

| Hyperparameter | Value |
| --- | --- |
| Grey-Scaling | True |
| Observation Size | 60x80 |
| Frames Stacked | 4 |
| Reward Clipping | False |

## C.2 ALGORITHM HYPERPARAMETERS

Table 7: Table showing the hyperparameters used in the **BTR** + RISE algorithm.

| Hyperparameter | Value |
| --- | --- |
| Learning Rate | 1e-4 |
| Discount Rate | 0.997 |
| N-Step | 3 |
| IQN Taus | 8 |
| IQN Number Cos' | 64 |
| Huber Loss $\kappa$ | 1.0 |
| Gradient Clipping Max Norm | 10 |
| Parallel Environments | 64 |
| Gradient Step Every | 64 Environment Steps (1 Vectorized Environment Step) |
| Replace Target Network Frequency (C) | 500 Gradient Steps (32K Environment Steps) |
| Batch Size | 256 |
| Total Replay Ratio | $\frac{1}{64}$ |
| Impala Width Scale | 2 |
| Spectral Normalization | All Convolutional Residual Layers |
| Adaptive Maxpooling Size | 6x6 |
| Linear Size (Per Dueling Layer) | 512 |
| Noisy Networks $\sigma$ | 0.5 |
| Activation Function | ReLu |
| $\epsilon$-greedy Start | 1.0 |
| $\epsilon$-greedy Decay | 8M Frames |
| $\epsilon$-greedy End | 0.01 |
| $\epsilon$-greedy disabled | 100M Frames |
| Replay Buffer Size | 1,048,576 Transitions ($2^{20}$) |
| Minimum Replay Size for Sampling | 200K Transitions |
| PER Alpha | 0.2 |
| Optimizer | Adam |
| Adam Epsilon Parameter | 1.95e-5 (equal to $\frac{0.005}{batchsize}$) |
| Adam $\beta1$ | 0.9 |
| Adam $\beta2$ | 0.999 |
| Munchausen Temperature $\tau$ | 0.03 |
| Munchausen Scaling Term $\alpha$ | 0.9 |
| Munchausen Clipping Value ($l_0$) | -1.0 |
| RISE Encoder | Pretrained ResNet18 (Penultimate Layer) |
| RISE Embedding Size | 512 |
| RISE Context Length | 160 |
| RISE LSTM Size | 256 |
| RISE Upscaling Linear Size | 2304 (BTR-Impala's Convolutional Output Size) |
| RISE Upscaling Layer Activation | Sigmoid |

Table 8: Table showing the hyperparameters used in the Vectorized **Rainbow** + RISE algorithm. We opted to use a similar vectorization scheme to BTR to significantly increase walltime compared to the original Rainbow DQN. Apart from hyperparameters changes and addition of RISE, this algorithm is identical to that of Hessel et al. (2018). Differences are highlighted in bold.

| Hyperparameter | Value |
|---|---|
| Learning Rate | **1e-4** |
| Discount Rate | 0.99 |
| N-Step | 3 |
| C51 Atoms | 51 |
| C51 Min/Max Values | [-10, 10] |
| Parallel Environments | **64** |
| Gradient Step Every | **64 Environment Steps** (1 Vectorized Environment Step) |
| Replace Target Network Frequency (C) | **500 Gradient Steps** (32K Environment Steps) |
| Batch Size | **256** |
| Total Replay Ratio | $\frac{1}{64}$ |
| Linear Size (Per Dueling Layer) | 512 |
| Noisy Networks $\sigma$ | 0.5 |
| $\epsilon$-greedy | Not Used |
| Activation Function | ReLu |
| Replay Buffer Size | 1,048,576 Transitions ($2^{20}$) |
| Minimum Replay Size for Sampling | 80K Transitions |
| PER Alpha | 0.5 |
| Optimizer | Adam |
| Adam Epsilon Parameter | **1.95e-5** (equal to $\frac{0.005}{batchsize}$) |
| Adam $\beta 1$ | 0.9 |
| Adam $\beta 2$ | 0.999 |
| Gradient Clipping Max Norm | 10 |
| RISE Encoder | Pretrained ResNet18 Image Classifier (Penultimate Layer) |
| RISE Embedding Size | 512 |
| RISE Context Length | 160 |
| RISE LSTM Size | 256 |
| RISE Upscaling Linear Size | 3136 (Rainbow's CNN Convolutional Output Size) |
| RISE Upscaling Layer Activation | Sigmoid |

## D   LSTM Hidden Size

One hyperparameter which is closely related to RISE's implementation is the hidden state dimension of the LSTM layer. In Figure 30, we test different widths, finding this hyperparameter to be largely unimportant, with all tested values giving similar performance.

## E   Architecture Diagrams

Here we show architecture diagrams for BTR + RISE and Rainbow + RISE, shown in Figures 31 and 32 respectively.

While running LSTM models to compare RISE against, we tested two different architectures, demonstrated in Figures 34 and 33. The former was largely inspired by that of Badia et al. (2020), however we found this architecture to give poor performance. Instead, we opted to use a similar architecture to that of RISE, and found this to be a significant improvement. For a comparison of results, see Figure 35.

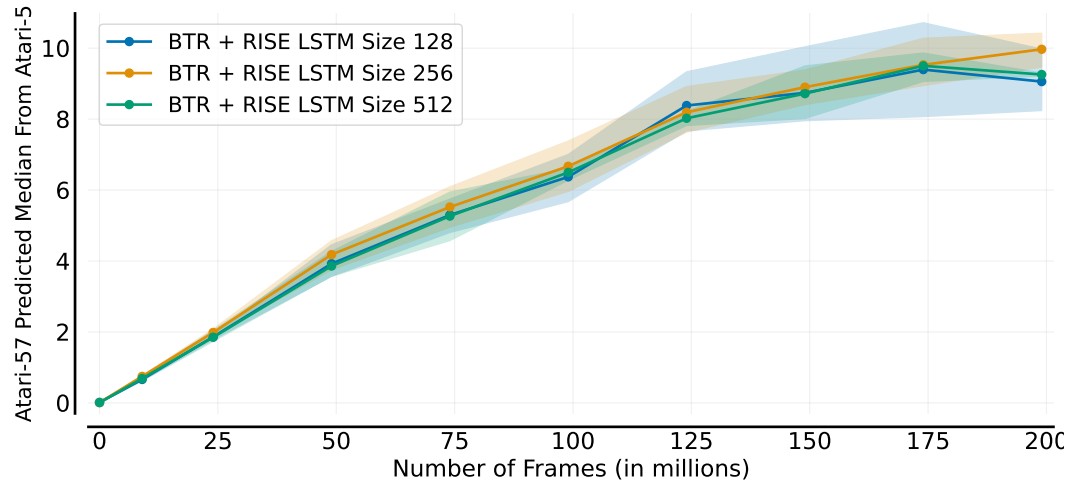

Figure 30: Performance of BTR with our proposed RISE framework, using a different widths of the Long-Short Term Memory (LSTM) layer. Results are averaged over 3 seeds, with shaded areas showing 95% confidence intervals. Throughout the paper, a width of 256 is used unless otherwise stated.

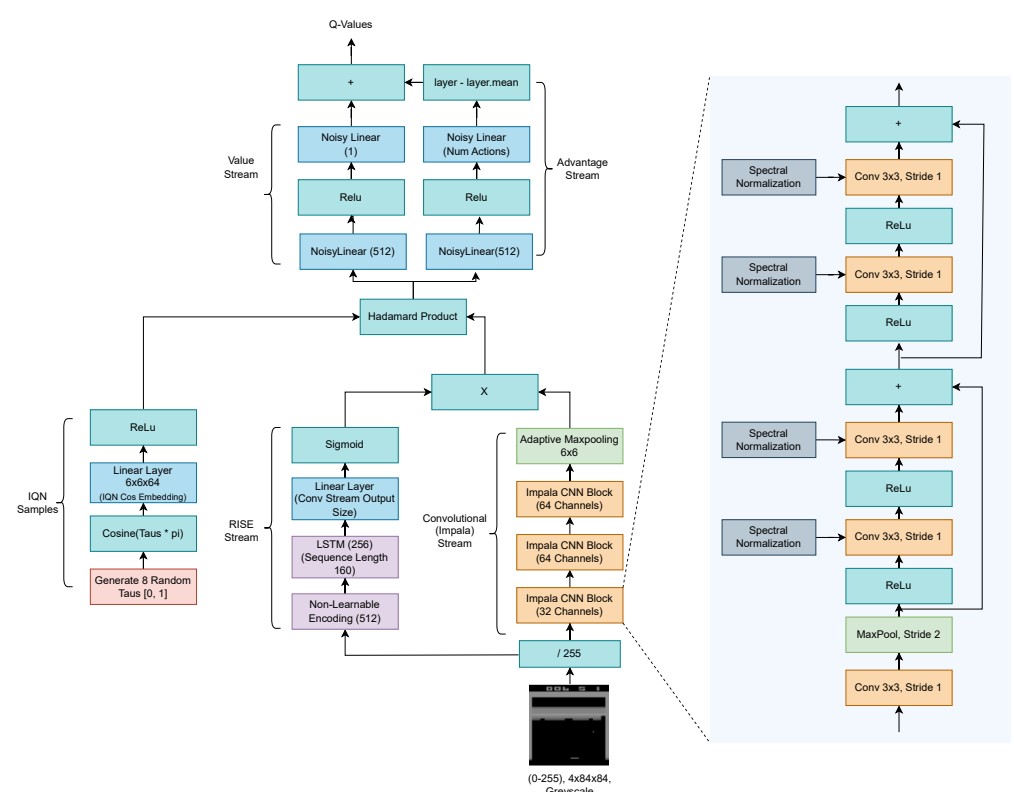

Figure 31: Detailed architecture diagram of the BTR + RISE model.

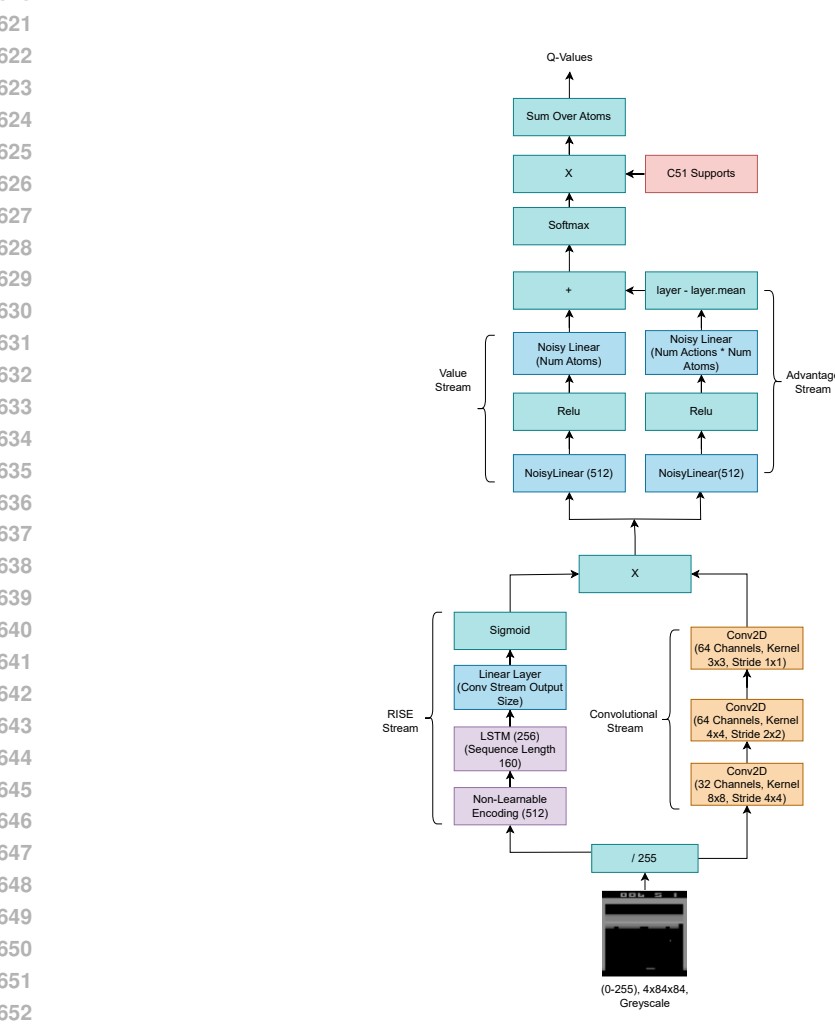

Figure 32: Detailed architecture diagram of the Rainbow + RISE model.

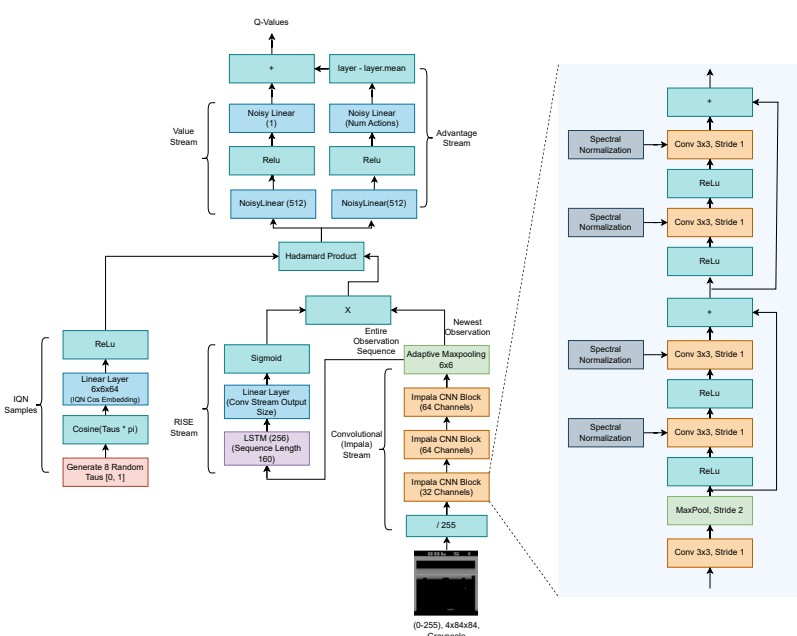

Figure 33: Detailed architecture diagram of the BTR + LSTM model, using a architecture style similar to RISE.

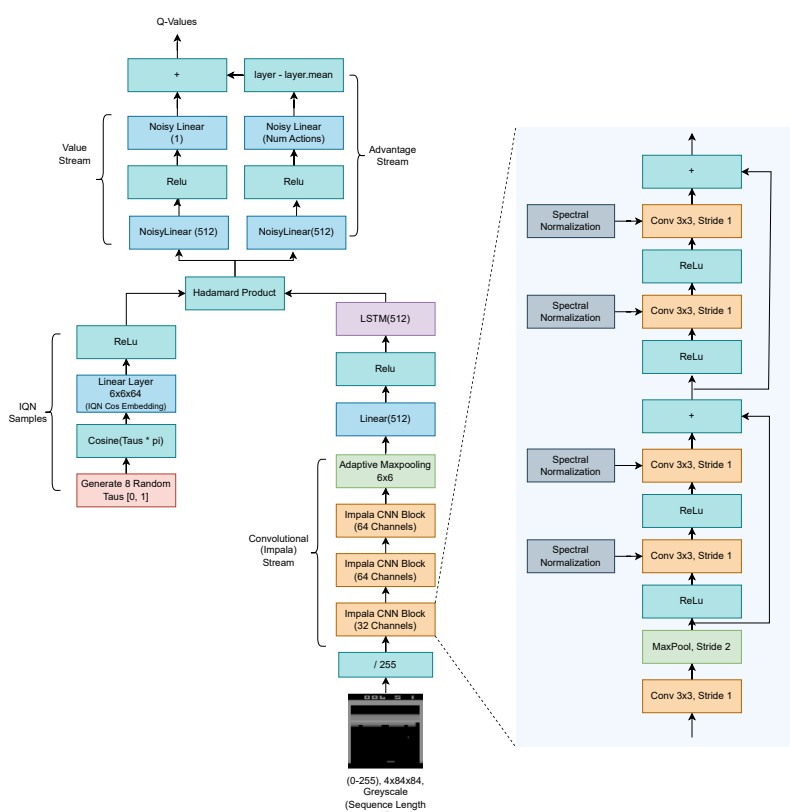

Figure 34: Detailed architecture diagram of the BTR + LSTM model. Design choices for this architecture were largely inspired by Badia et al. (2020).

## F  RISE IMPLEMENTATION DETAILS

One small yet important implementation detail of RISE is how observations are processed before being passed into the pretrained vision encoder. Since BTR uses the standard 4-framestack of 84x84 greyscale images, we convert these to match the data the vision encoder was trained on (for the ResNet18 model, this is a RGB 224x224 image). We do this by simply upscaling the image fed to BTR, and applying the same color normalization scheme used by ImageNet (the dataset the model was trained on). Furthermore, we use the most recent single frame from the BTR observation framestack. We deemed it unnecessary to use a framestack for the LSTM which was already fed a sequence of frames. On the other hand, we opted to continue using the 4-framestack for the BTR encoder. This still allows the model to quickly learn things such as motion by using the difference between frames. Additionally, the computational burden between using 4 frames and a single frame isn't that significant since it only effects the first CNN block (although does make the overall implementation a little more complex).

## G  EVALUATION DETAILS

Throughout all experiments used in this research, we opted to use separate evaluator processes rather than using a 100 episode evaluation as used in Machado et al. (2018). Running the full evaluations was far beyond our computational budget due to the length of the evaluations. On many games, BTR + RISE was able to use the full 108,000 frames (max time on the Atari environment) from a very early stage in training. This meant the time taken to perform a single run of BTR + RISE would go from

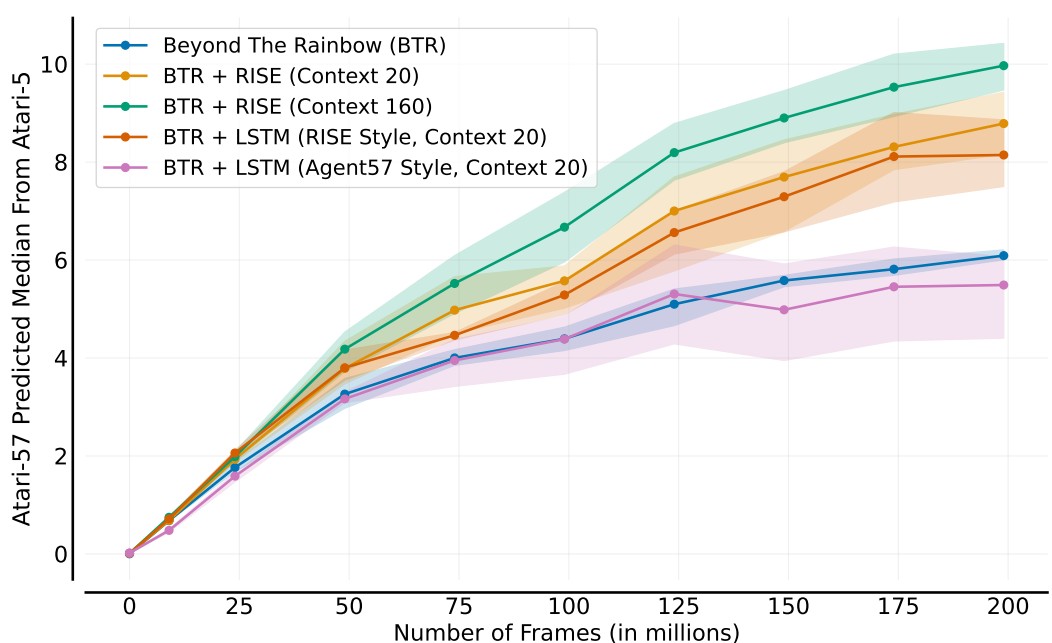

Figure 35: Comparison of different LSTM architectures added to BTR. Shaded areas show 95% confidence intervals across 3 seeds.

23 hours to over 120 hours (even when training/evaluating in parallel) in some cases (particularly Atlantis, ElevatorAction and Phoenix), beyond the computational resources we have access to.

## H  RECURRENT PARADIGM DETAILS

Kapturowski et al. (2018) showed that the techniques stored-state and burn-in can be used to tackle the inaccurate hidden-state problem in off-policy recurrent RL. While RISE already uses a burn-in-like mechanic with its context length, it can still utilize stored states to improve the accuracy of hidden states. Furthermore, it is also possible to burn-in observations before those being learned on to produce more accurate hidden states (observations prior to the context length can be passed through the LSTM without gradients). This differs from the burn-in used in R2D2, which does not indicate whether gradients are used or not. We will call this method no-grad-burn-in. The standard method presented throughout this paper uses a context length of 160, acting similarly to a burn-in period. In Figure 36, we find that other methods had little impact on performance on Atari-5.

## I  COMPUTE RESOURCES

Our training was performed using a variety of different hardware, listed below with the walltime for BTR + RISE (context length 160) for 200M Atari frames:

- NVIDIA H100 clusters, 48 CPU cores, 500GB memory (27 Hours)
- NVIDIA A100 clusters, 48 CPU cores, 500GB memory (29 Hours)
- NVIDIA L4 clusters, 12 CPU cores, 1TB memory (51 Hours)
- NVIDIA Tesla V100 clusters, 48 Cores, 500GB memory (57 Hours)
- NVIDIA RTX Quadro 8000 clusters, 48 Cores, 500GB memory (70 Hours)
- NVIDIA RTX 4090, Intel-i9 13900K (24 Cores), 64GB memory - Desktop PC (20 Hours)

Despite using compute clusters, we do not perform any form of distributed training, in contrast to many other high-performance RL algorithms (Kapturowski et al., 2018). Furthermore, while we had a large amount of memory available, usage never exceeded 30GB.

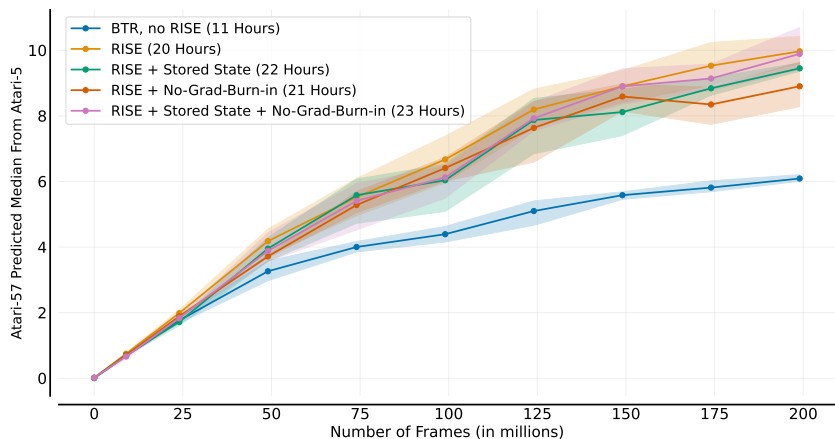

Figure 36: Analysis of BTR with our proposed RISE framework, with varying methods for dealing with the recurrent off-policy learning problem on Atari-5. All runs used 3 seeds, with shaded areas showing 95% confidence intervals. Legend shows walltime on a desktop PC with an RTX4090. A context length of 160 was used, with a no-grad-burn-in of 80 where relevant.

Total Experiments Conducted (all used 3 seeds, time given in hours):
BTR + RISE (Atari-57): 57 * 3 * 27 = 4617 (H100)
Vectorized Rainbow (Atari-5): 5 * 3 * 17 = 255 (RTX Quadro 8000)
Vectorized Rainbow + RISE (Atari-5): 5 * 3 * 22 = 330 (L4)
BTR-LSTM (Atari-5): 5 * 3 * 150 = 2250 (H100)
Rainbow-LSTM (Atari-5): 5 * 3 * 50 = 690 (RTX 4090)
Procgen (16 Tasks): 16 * 3 * 32 = 1536 (Tesla V100)
VizDoom (7 Tasks): 7 * 3 * 20 = 420 (A100)
Context Length Ablation (Atari-10): 10 * 3 * (16 + 17 + 18) = 1530 (RTX 4090)
Recurrency Paradigm (Atari-5): 5 * 3 * (27 + 30 + 28 + 31) = 1740 (H100)
Sequence Architecture (Atari-5): 5 * 3 * (38 + 20 + 27) = 1275 (RTX 4090)
LSTM Size (Atari-5): 5 * 3 * 29 = 435 (A100)
BTR with Agent57-style LSTM (Atari-5): 5 * 3 * 150 = 2250 (H100)
No Learnable Encoder (Atari-5, 1 seed): 5 * 9 = 45 (L4)

Total GPU Hours: 17373

## J WHY RECURRENT MODELS BENEFIT PERFORMANCE

Videos are available of BTR-RISE playing Atari-5 at the anonymous link: https://www.youtube.com/playlist?list=PL0BaIoEl7EKBfxRJnIXci3gVCZwftEhtb.

To understand what the LSTM is being used to do, below we have compiled a list of environments, and how the trained policies of BTR and BTR-RISE differ. Some of these include timestamps from the linked videos.

Atari Venture - Venture involves moving to many different rooms from a map screen and then acquiring an artifact. Obtaining a single reward requires taking hundreds of timesteps. This task, however, is not a particularly hard exploration task; random policies can sometimes reach rewards. BTR can reach rewards during exploration, but is unable to learn to repeat the behaviour, whereas BTR-RISE can. This suggests that BTR-RISE was able to do better long-term credit assignment and learn from the rewards it was given.

Atari NameThisGame - This game involves a deep-sea diver shooting an octopus, and periodically being given oxygen to breathe by a fisherman. The BTR agent often misses oxygen, as it doesn't know when the oxygen was last given. The BTR-RISE agent leaves the fisherman after getting oxygen, then returns to them before oxygen is given again (this can be seen at 12:30 in the video).

Atari Qbert - This game involves a player moving around to flip coloured tiles while avoiding an enemy. One of the later levels contains an exploit, where the player can repeatedly flip the same tile for more points. Since the game's enemy moves predictably, BTR-RISE can predict the enemy's moves, allowing it to exploit the game (see 6:30 in the video). BTR struggles to avoid the enemy due to this lack of prediction.

Atari BattleZone, Phoenix -These games give the player multiple lives, with the agent only receiving a terminal after the final life is lost (the agent does not receive any negative reward or signal for losing intermediate lives). The BTR agent is extremely careless with early lives, then plays much more safely when it only has one life remaining. Conversely, the BTR-RISE agent is much more careful with early lives, indicating better long-term credit assignment.

Procgen Coinrun - This environment has the agent try to reach a coin, often taking many steps to reach the coin. The BTR agent is sometimes able to reach the coin via random exploration, but fails to learn from those signals to reach the coin consistently. BTR-RISE, on the other hand, learns the behaviour quickly.

VizDoom HealthGathering - This task has the player run around a simple room with a first-person camera, picking up health packages. The BTR agent often gets stuck, moving its camera side-to-side repeatedly when it cannot see any packages (effectively, the agent alternates between viewing two locations, neither of which has packages). BTR-RISE does not get stuck in this way, likely because it has the previous location within its context length.

Miniworld environments - We found that even BTR without RISE was able to solve many of the tasks (Hallway, WallGap, YMaze, TMaze) despite not using a recurrent model; however, RISE was able to aid in the remaining environments (One Room, Four Rooms, Sidewalk). Both BTR and BTR + RISE failed in ThreeRooms due to a lack of exploration.

