# OpenReview forum: "Recurrent Off-Policy Deep Reinforcement Learning Doesn’t Have to be Slow"
_ICLR.cc/2026/Conference — Submitted to ICLR 2026_

### Official Review · Reviewer_dZ14 · 2025-10-28

**Soundness:** 4
**Presentation:** 4
**Contribution:** 3
**Rating:** 6
**Confidence:** 3

**Summary:**

The paper introduces RISE (Recurrent Integration via Simplified Encodings) a framework for integrating recurrent neural networks efficiently into off-policy reinforcement learning without incurring the heavy computational cost typical of recurrent methods. RISE separates the encoding of recent and long-term observations by combining learnable encoders for immediate inputs and non-learnable (pretrained) encoders for historical context. This enables recurrent models to leverage temporal dependencies without redundant convolutional passes. When applied to strong baselines like Beyond The Rainbow (BTR), RISE achieves up to 35.6% improvement in human-normalized interquartile mean performance on the Atari benchmark while drastically reducing walltime. The paper validates RISE across Atari, Procgen, VizDoom, and Miniworld.

**Strengths:**

This paper presents a original contribution by removing one of the key computational bottlenecks in recurrent off-policy reinforcement learning through its RISE framework, which cleverly combines learnable and non-learnable encoders to enable efficient use of recurrence. The quality of the work is strong, supported by rigorous experimentation across multiple benchmarks and careful ablation studies that validate the design choices. The paper is clearly written and well-structured, effectively communicating both the motivation and the technical details of the approach. Its significance is substantial, as RISE democratizes access to high-performing recurrent off-policy RL methods by drastically reducing computational demands, potentially broadening their adoption and impact in both academic and applied research contexts.

**Weaknesses:**

While the paper makes a strong contribution, several areas could be improved. First, the analysis of temporal credit assignment and how RISE’s separation of encoders affects long-horizon dependencies is limited; more diagnostic experiments (e.g., on tasks requiring extended memory like DMLab or Meta-World) would clarify this. Second, the ablation studies could go deeper by isolating the effects of encoder pretraining and architectural design choices. Finally, the theoretical motivation for the hybrid encoder structure remains mostly empirical—adding a formal analysis or simplified toy model could further solidify the conceptual contribution.

**Questions:**

1.Could the authors elaborate on the specific mechanism of integration between the learnable and non-learnable encoders? For instance, how is the representation from the non-learnable encoder normalized or aligned with that from the learnable one before fusion?

2.The method’s empirical success is clear, but the theoretical reasoning for why the hybrid encoder structure preserves recurrent credit assignment efficiency is underdeveloped.

---

> ### Author Response · Authors · 2025-11-14
>
> Thank you for your thoughtful and detailed review, which will help us improve our work. We are especially grateful for your detailed strengths section, which effectively explains the impact our work will have in academic and applied research contexts, and applauds our rigorous experimentation and analysis. We will now systematically address each of the remaining concerns.
>
> **Weakness 1 (Long-term credit assignment):** Sadly, applying RISE to meta-world or DMLab within the rebuttal period may not be possible. However, we would like to raise a couple of points on this topic: Firstly, Appendix J has an analysis of the difference in policies between BTR and BTR + RISE, including provided videos with timestamps, many of which demonstrate increased ability to do long-term credit assignment. Specifically, in games with multiple lives, such as BattleZone and Phoenix, the BTR + RISE was far more careful with early lives, whereas the BTR agent would only be cautious with a single life remaining.
>
> **Weakness 2 (design choices):** Please see our response to question 1.
>
> **Weakness 3 (hybrid encoder structure):** The motivation for the hybrid structure was simply one of necessity -  since we needed to use two encoders (one for learnable and one for non-learnable features), this forced us to use separate encoders. Without this, we would suffer from the same issues as prior work (recurrent-state staleness, high compute costs). These issues were explored in detail in R2D2 [1].
>
> **Question 1 (Integration of the two encoders):** To combine the two representations, we take the following steps (please see Figure 2 for the diagram):
> After the output of the LSTM, this provides us with a 512-dimensional encoding. In order to match the shape of this encoding with the learnable encoder’s output, we upscale this encoding using a linear layer, with 512 inputs and 2304 outputs (the output size of BTR’s learnable encoder). Before combining these via multiplication, we first apply a sigmoid function to the non-learnable stream. This acts as an activation function following the linear layer; however, we chose to use a sigmoid to mimic an attention mechanism, whereby the non-learnable stream can “turn off” parts of the learnable stream. Despite this, we hypothesize that any activation (such as relu) may provide similar performance. We did not perform an extensive ablation on different ways this combination could be done; for example, using a different activation function, adding/averaging the two streams, or simply concatenating the LSTM’s output with the learnable encoder’s output are all possible. We will however be adding many of these ablations in an Appendix to our revised paper.
>
> **Question 2 (Reasoning for improved long-term credit assignment):** We don’t believe that the hybrid encoder structure itself provides better long-term credit assignment, but rather that all recurrent models do. This idea came from the Agent57 algorithm [2], which claims and empirically demonstrates that increasing the backprop through time window improves long-term credit assignment. In this work, as explained in their Appendix H6, Solaris achieved the largest performance improvement and was a long-term credit assignment problem. The agent needs to navigate a grid to seek out enemies, which only many timesteps later provides a reward. Our results mirror this, also observing a large improvement in Solaris, with BTR achieving no positive reward. In line 31 of the paper, we state that recurrent models (not necessarily hybrid architectures) can be used to overcome long-term credit assignment.
>
> **Conclusion:** We hope we have clarified the purpose of the dual-stream architecture and the ideas behind improved long-term credit assignment. While the black-box nature of neural networks makes it difficult to isolate the impact of these design choices, we are happy to run some additional ablations regarding the combination of the two streams, including concatenation and addition instead of multiplication, and also combining the two streams before the LSTM rather than after. We hope that these additional clarifications and experiments will encourage you to advocate for acceptance, or increase your score.
>
> **References:**
>
> [1] Kapturowski, Steven, et al. "Recurrent experience replay in distributed reinforcement learning." International conference on learning representations. 2018.
>
> [2] Badia, Adrià Puigdomènech, et al. "Agent57: Outperforming the atari human benchmark." International conference on machine learning. PMLR, 2020.

---

> > ### Comment · Reviewer_dZ14 · 2025-11-27
> >
> > **We appreciate the authors’ response. I have read it carefully and decided to keep my original rating.**

---

### Official Review · Reviewer_WuVd · 2025-10-30

**Soundness:** 3
**Presentation:** 3
**Contribution:** 3
**Rating:** 6
**Confidence:** 2

**Summary:**

This work proposes a novel method, RISE, to make recurrent models computationally efficient in the off-policy RL domain. In short, it proposes to significantly save on computational resources by combining a standard (trained) encoder to process the current state input (image) with a fixed non-learned encoder to process pre-computable embeddings for each state in the preceding part of the sequence. These (cheaper) embeddings are stored in the replay buffer and fed to the recurrent model during off-policy training. This approach provides the full benefits of long-term context without the expensive re-computation which is often induced by processing every past frame of a trajectory, typically done with an expensive CNN encoder. Ultimately this approach results in large wall-clock time speedups and near state-of-the-art absolute performance in a range of RL tasks.

Note that this specific area is not my expertise, hence my low confidence score for this review, though I read the paper thoroughly nonetheless.

**Strengths:**

- The proposed solution for reducing computational cost in this work is (as far as I know) original and has a significant impact in reducing computational cost and in broadening the computational accessibility of RL modelling
- Experimental validation are carried out across a range and diverse set of tasks and with comparisons against state of the art baselines - a significant undertaking
- Ablation studies and studies of what affects the performance of the fixed embedding are well carried out
- The use of such a setup is compatible with a range of RL learning methods and is therefore of general interest and potential impact

**Weaknesses:**

- Though training time is reduced by use of this method, the peak memory usage required for this method is likely to be higher than otherwise when taking into consideration the model used to produce embeddings
- The limitations of this method are likely heavily linked to any limitations in the non-learnable encoder0 This likely means that for tasks which are not visual in nature, or have states which are OOD for the (fixed) embedding network, the method would likely fail. This may also be the reason why this method fails for some tasks over others
- Theoretically, the contributions of this work are rather limited and weak. The RL tasks, or RL process in general is not better understood as a result of this paper and it remains somewhat unclear how much the benefits seen are due to the embedding-producing network

**Questions:**

- The limitations of this work are relatively little discussed in the space of what types of tasks (in vs out of distribution for the non-learnable encoder) this approach would work for? How could one make smart choices about which encoders to use for embeddings?

I don't have many questions as this work seems rather comprehensive in nature. Note that my familiarity with such RL work is rather minimal. Overall, this seems like a very promising addition to RL methods.

---

> ### Author Response · Authors · 2025-11-14
>
> Thank you for your detailed review, constructive feedback and for acknowledging the potential impact of our work and the rigor of our evaluation and analyses.
>
> **Weakness 1 (Memory Usage):** Yes, peak memory usage will be slightly increased. However, storing a ResNet-18 on GPU is very small by today’s standards, using only around 47MB for its weights (11.4 million parameters * 4 bytes for float 32). Since the model is only used for inference, many of the typical costs (such as gradients) are not relevant. Even the feature maps computed during inference for each environment (64 in the case of BTR) only use around ~70MB. As mentioned on line 339, our method uses an additional 2GB of RAM to store the encodings, for a buffer size of 1M, which we also deem minor by today’s standards (this value is the same, regardless of context length).  We will add this information to the main body of the paper, in the same place we discuss the additional RAM costs.
>
> **Weakness 2 (Limitations of a Fixed Encoder):** Yes, it is a limitation of our method that the non-learnable encoder may not be able to capture the relevant features. We will include the following 2 sentences in Section 7 to acknowledge this limitation: “We acknowledge that the use of a fixed, pre-trained encoder may limit the model’s ability to extract useful features in some environments. In this sense, RISE acts as a trade-off, allowing greater compute-efficiency in exchange for adaptability if the encoder does not capture the relevant features.” It is worth noting that the features produced by trained, Deep CNNs tend to be very general [1], and are still likely to be useful in the majority of cases. We would also like to point out that the tasks we test on (such as Atari and Procgen) are a very different distribution from that of ImageNet, yet still provides a substantial improvement (as mentioned on line 379). As shown in Figure 9, even simply downsampling and flattening the image still provides a significant benefit. This may be the reason why it does not improve performance on some tasks; however, this may also be due to tasks not having partial observability or long-term credit assignment. Components of RL algorithms having selective improvements on some environments is extremely common; please see Figure 4 in Rainbow DQN [2] for an example of this.
>
> **Weakness 3 (Theoretical Contribution):** Yes, this is a fair critique as we agree our work focuses on empirical and practical improvement. Our Figure 9 looks to analyse the contribution of the embedding-producing network. We especially think that using downsampling rather than a pre-trained network was a reasonable attempt to shed light on this contribution, indicating that our method can be applied and achieve significant performance gains, even when pre-trained models are unavailable.
>
> **Question 1 (Distribution of Fixed Encoder):** As mentioned, many of the tasks we evaluate on are out of the distribution of the what the pre-trained encoder was trained on, indicating that the choice of pre-trained encoder may be relatively robust. Furthermore, we did not see significant differences between different pre-trained encoders. To best choose which encoder to use, we would simply recommend using an encoder with data trained on as similar a task as possible. For example, if videos of a task exist, it could be possible to train a new encoder on similar data, which may yield better performance.
>
> **Conclusion:** We believe we have rigorously addressed your outstanding concerns, and hope you will continue to advocate for the acceptance of our paper, or consider raising your score.
>
> **References:**
>
> [1] Babaiee, Zahra, et al. "The Master Key Filters Hypothesis: Deep Filters Are General." Proceedings of the AAAI Conference on Artificial Intelligence. Vol. 39. No. 2. 2025.
>
> [2] Hessel, Matteo, et al. "Rainbow: Combining improvements in deep reinforcement learning." Proceedings of the AAAI conference on artificial intelligence. Vol. 32. No. 1. 2018.

---

> > ### Comment · Reviewer_WuVd · 2025-11-27
> >
> > Thank you for your responses, I appreciate reading your perspective. The responses are helpful but do not move me to significantly reconsider my rating. Best of luck.

---

### Official Review · Reviewer_Wzuf · 2025-11-01

**Soundness:** 1
**Presentation:** 3
**Contribution:** 1
**Rating:** 2
**Confidence:** 5

**Summary:**

The paper targets an important but under-served practical problem in deep RL: recurrent off-policy agents are known to help in partially observable visual tasks, but they are often too expensive to train because every replayed sequence must be re-encoded frame-by-frame by a heavy vision backbone. The authors propose a simple two-stream design. For the long history/context part of the sequence, they precompute image features with a fixed encoder at data-collection time and store those features directly in the replay buffer. At training time, the recurrent module (LSTM) consumes only these precomputed features, so no CNN forward pass is needed for the historical frames. For the current timestep, they still use a normal, learnable vision encoder, and they fuse its output with the LSTM output to produce Q-values (or the usual off-policy head). This reduces the number of vision forward passes during training to something close to non-recurrent agents, while still allowing the agent to benefit from long temporal context. Empirically, plugging this into strong image-based off-policy baselines  improves human-normalized scores on Atari and other visual benchmarks under limited compute.

**Strengths:**

1. Well-motivated practicality. The paper correctly identifies a real bottleneck in recurrent off-policy RL: sequence-based replay plus heavy visual backbones is costly, so many practitioners simply do not turn on recurrence even when tasks are partially observable. The paper gives a plausible solution to this exact bottleneck.
2. Simple, reproducible mechanism. The main trick—precompute visual features for history and store them in replay—is conceptually simple and easy to reimplement in existing off-policy codebases.
3. Good empirical story. The experiments show that (i) you can retain the benefits of long context (up to ~160 steps) and (ii) you do not have to pay the usual multiplicative cost in CNN passes. Results on Atari and other visual / partially observable tasks support the claim that “recurrent off-policy is actually usable” under their design.
4. Compatibility with strong baselines. The method is evaluated by adding it to competitive off-policy agents (not by introducing a weak agent and claiming gains). This makes the numbers more trustworthy.
5. Ablations are informative. The paper does some ablation on fixed-encoder choice and context length, which helps clarify when the method helps more (tasks with partial observability, longer-horizon credit, more visual complexity).

**Weaknesses:**

1. Incremental from an RL perspective. The core contribution is an architectural / systems rearrangement—moving expensive vision to data-collection time and having the LSTM consume precomputed features—not a new principle for off-policy RL. The underlying learning rule, off-policy correction, replay usage, and handling of partial observability all stay standard. This makes the contribution feel more like “making an existing recipe cheaper” than “a new RL idea.”
2. No deeper treatment of recurrent off-policy issues. Recurrent off-policy is known to have subtleties (e.g. mismatch between behavior-policy hidden state and learner hidden state, burn-in choices, bias due to truncated sequences). The paper does not propose new corrections or analyses for these; it just makes the pipeline faster. This limits conceptual novelty.
3. Reliance on fixed visual features is under-analyzed. Because the LSTM consumes non-learnable features for most of the context, the representation seen by the recurrent module is partially decoupled from the task loss. The paper shows it works empirically, but it does not dig into when a fixed encoder might become a bottleneck (e.g. domains very different from the pretraining source, tasks that need fine-grained spatial cues across time).
4. Performance improvements could be framed as “better compute–data trade.” Many of the reported gains can be interpreted as: “we could finally afford longer context and bigger batches because it was cheap enough,” rather than “this specific architecture is inherently better.”

**Questions:**

1. How sensitive is the method to the exact fixed encoder? If we swap ResNet-18 pretrained on ImageNet with a much lighter conv stack (e.g. 4-layer Atari-style conv), do we still see the same benefits on Atari, or does the LSTM stop being useful because the features are too weak?
2. What is the storage overhead for keeping precomputed features for long contexts in the replay buffer? For very long contexts or higher-res observations, does this become a memory bottleneck?
3. Can the authors report results where both the historical encoder and the current encoder are learnable but updated at different frequencies (e.g. slow EMA for the history encoder) so we can see the trade-off between flexibility and cost?
4. For tasks with non-visual partial observability (e.g. missing proprioception, delayed rewards), would the method still help, or is it mostly an image-RL trick?

---

> ### Author Response · Authors · 2025-11-14
>
> We would like to express our gratitude for your thorough review and insightful comments. We appreciate your applause of our work, including our motivations, simplicity, story, compatibility with baselines, and ablations. We will now rigorously address each of your concerns.
>
> **Weakness 1 (Incremental Improvement):** Yes, our work does focus on making an existing idea cheaper; however, we do not believe this is insignificant. As stated in our introduction, the highest-performing algorithms are off-policy recurrent models, which take significant compute resources. This leads to only a few well-funded labs being able to contribute to the field, whereas our work has the potential to make high-performance RL widely accessible, therefore rapidly accelerating progress. Furthermore, this has positive environmental impacts, and allows those with more computing power to iterate on ideas significantly faster.
> Other ideas, such as distributed RL [1, 2] and end-to-end GPU training [3], were largely focused on “making an existing recipe cheaper” and were some of the most significant works in RL. For more information on the problems caused in RL by computational costs, please read [4]. Many of the strengths listed in your review support this argument - you mention that our approach is “well-motivated” and has a “good empirical story”. Given your comments, it seems that you agree that this problem is a major bottleneck for many researchers. Furthermore, you mention that our solution provides a simplistic way to mitigate that problem while maintaining strong performance, which has never been proposed before.
>
> **Weakness 2 (Recurrent Off-Policy Issues):** Please see Appendix H, including Figure 19, where we have a detailed ablation and discussion on burn-in and stored-state. This appendix is referenced in the main paper on line 195. While we don’t propose a new correction, our method makes it substantially faster to use longer context lengths, which itself leads to more accurate hidden states. Furthermore, since encoder outputs don’t change, the error from staleless is reduced.
>
> **Weakness 3 (Fixed Visual Features):** We agree that, in theory, the pre-trained encoder may become problematic for tasks that require fine-grained spatial cues. We are happy to add 2 sentences in Section 7 to acknowledge this limitation, saying “We acknowledge that the use of a fixed, pre-trained encoder may limit the model’s ability to extract useful features in some environments. In this sense, RISE acts as a trade-off, allowing greater compute-efficiency in exchange for adaptability if the encoder does not capture the relevant features.” It is worth noting that the features produced by trained, Deep CNNs tend to be very general [5], and are still likely to be useful in the majority of cases. While this is a valid limitation of our work, you provide the comment “domains very different from the pretraining source”. We’d like to point out that our pretrained encoder was trained on ImageNet, and none of our benchmarks resembled ImageNet. This means our evaluation already includes environments with a substantial domain shift, and still provide significant performance benefits (this is mentioned on line 379).
>
> **Weakness 4 (Better Compute-Data Trade):** We use the same batch size as Beyond The Rainbow, meaning none of our gains come from a larger batch size (we update the same number of states per gradient step, and perform the same number of gradient steps). We agree that many of the gains do come from previously not being able to use long context lengths due to computational restrictions. Our work will mean that many other researchers will also be able to use longer context lengths and see improved performance with minor compute increases. Furthermore, Figure 31 shows that even using a typical LSTM paradigm where all observations in the context length are recomputed during gradient steps, we find that a RISE-style architecture still outperforms existing architectures substantially when applied to BTR.

---

> > ### Author Response · Authors · 2025-11-14
> >
> > **Question 1 (Sensitivity to the Encoder):** Please see Figure 9, as we already have ablated different methods of producing encodings. This includes simply downsampling and flattening the observations, and we find that this still provides a substantial performance improvement, indicating that the LSTM is still useful in many environments. Also, we’d like to point out that by modern standards, performing only inference on a ResNet-18 once per sample is not particularly compute-heavy.
> >
> > **Question 2 (Storage Overhead):** As mentioned on line 339, our method uses just 2 GB of additional memory, even for a replay buffer size of 1M. Furthermore, this memory size does not change based on the context length. We store exactly one encoding per observation, regardless of the context length. All observations and their encodings are stored together, meaning when an observation is sampled, we can simply look back through the episode and fetch the recent history of encodings, with minimal memory/walltime costs.
> >
> > **Question 3 (Two Learnable Encoders):** This is a very interesting idea which we believe will improve our paper, for which we thank you. We are happy to test a second version of RISE, where the historical encoder uses a slow EMA of the main encoder network. This will introduce staleness into the encodings in exchange for the encodings being adaptable. Also, as mentioned in line 476, the network may benefit from the knowledge provided by a pre-trained model, which may be lost. We agree that this is an interesting experiment, which we will report on the Atari-5 benchmark using 3 seeds before the end of the rebuttal period.
> >
> > **Question 4 (Domains of Application):** While we primarily explore the image-based setting, as discussed in line 472, this work can be applied to other domains, such as CNNs for audio data, or transformers for text-based tasks (this may have a heavy compute cost, but may provide the model with useful knowledge). Even tasks that simply use a vector of numbers could apply our method; however, they would not see significant walltime decreases if encoder architectures are less expensive.
> >
> > **Conclusion:** We believe we have thoroughly addressed each of your concerns, especially emphasizing the importance of reducing the computational burden of RL, and are currently working to produce an additional experiment as per your suggestion. We sincerely hope you are willing to reconsider your score based on this information.
> >
> > **References:**
> >
> > [1] Vinyals, Oriol, et al. "Grandmaster level in StarCraft II using multi-agent reinforcement learning." nature 575.7782 (2019): 350-354.
> >
> > [2] Berner, Christopher, et al. "Dota 2 with large scale deep reinforcement learning." arXiv preprint arXiv:1912.06680 (2019).
> >
> > [3] Gallici, Matteo, et al. "Simplifying Deep Temporal Difference Learning." The Thirteenth International Conference on Learning Representations.
> >
> > [4] Ceron, Johan Samir Obando, and Pablo Samuel Castro. "Revisiting rainbow: Promoting more insightful and inclusive deep reinforcement learning research." International Conference on Machine Learning. PMLR, 2021.
> >
> > [5] Babaiee, Zahra, et al. "The Master Key Filters Hypothesis: Deep Filters Are General." Proceedings of the AAAI Conference on Artificial Intelligence. Vol. 39. No. 2. 2025.

---

> > > ### Author Response · Authors · 2025-11-26
> > > **Please see updated version**
> > >
> > > Thank you again for your review.
> > >
> > > During the rebuttal period, we have rigorously addressed all concerns raised in your original review and provided the experiments you requested, as well as additional ones, in our revised PDF. This includes new experiments using stale encodings from the main network encoder for the LSTM, both with a duplicate model and using an EMA as per your suggestion. Furthermore, in response to other reviewers, we added entirely new ablations, including the architectural design decisions used in RISE, and a more detailed comparison to R2D2.
> > >
> > > We sincerely ask that you review our answers to your questions, our list of changes, and the revised PDF, as we believe we have made highly significant changes that may impact your decision.

---

### Official Review · Reviewer_VpvB · 2025-11-01

**Soundness:** 3
**Presentation:** 3
**Contribution:** 2
**Rating:** 4
**Confidence:** 4

**Summary:**

This paper investigates how to improve the computational efficiency of recurrent off-policy image-based deep reinforcement learning methods by replacing the image encoder learned from scratch by using pretrained vision encoder like ResNet when encoding the historical images for Q-value computation. Experimental results show that their method outperform non-recurrent methods, achieve similar performance as existing recurrent methods while being more computationally efficient.

**Strengths:**

1. The paper is clearly motivated and the mothodology is clearly explained.
2. The method is thoroughly evaluated on different benchmarks.

**Weaknesses:**

1. The novelty of the paper seems limited to me, as the key contribution is to use frozen pretrained vision encoder instead of learning a new vision encoder from scratch to reduce computational cost. The idea of utilizing the strong prior of pretrained models for different downstream tasks has been extensively explored in different domains, and it's not too surprising to me that it will also work for off-policy RL on commonly used RL benchmarks with relatively simple visual features.
2. MEME and Dreamer-v3 perform better than the proposed method overall. Why not also apply your method to them to see if they can work better or faster? Is it due to limited computational budget?

**Questions:**

1. Why integrating recurrent models has minimal computational cost for on-policy RL but not for off-policy RL?
2. Why not also use the pretrained vision encoder for the current observation?

---

> ### Author Response · Authors · 2025-11-14
>
> Thank you for your detailed review, and for applauding our motivations and analyses. We will now thoroughly discuss all of the outstanding issues raised:
>
> **Weakness 1 (Clarification of Novelty):** While the core idea of using a pretrained vision model has been explored extensively in other domains, we believe our novelty extends beyond this. Firstly, as shown in Figure 9, even when not using a pretrained model (using downsampling), our approach still improves performance to a significant margin (an IQM increase from 6 to 7 with non-overlapping 95% confidence intervals). While our approach benefits from pretrained models, we believe this provides strong evidence that our improved performance is not solely based on this. Furthermore, our dual-stream approach of using both a learnable and non-learnable encoder is far less common in the literature. While our approach does provide a strong prior from a pre-trained model, which can help performance, the key innovation is to enable the use of recurrent models for off-policy RL without drastically increasing compute costs.
>
> **Weakness 2 (Application to MEME and Dreamer-v3):** We cannot apply RISE to MEME as the code for this algorithm was never made open source, and would be an extremely large undertaking to produce an accurate re-implementation of such a complicated algorithm with limited available details. While applying RISE’s style to Dreamer-v3 is possible, this change would not be straightforward. Dreamer-v3’s actor and critic learn from the abstract representations learned by the world model, which itself uses the gradients from the actor and critic. Applying RISE to a model-based algorithm like Dreamer-v3 is possible, it would be a significant undertaking beyond the scope of this paper to deal with the intricacies that may arise from this paradigm shift. Additionally, as you mention, both of these algorithms also have large compute requirements - given that Dreamer-v3 uses a large world-model, even with RISE this may still have a substantial compute cost.
>
> **Question 1 (Difference between recurrent on and off-policy RL):** The major difficulty in applying recurrent models to off-policy RL arises from the changing encoder output. In an on-policy algorithm, when acting, we produce the up-to-date encoder output and hidden states, which can be used to perform gradient updates. In contrast, in off-policy RL, suppose we sample some transition from our replay buffer which was collected many gradient steps ago. We need to compute the encoder’s output for all observations in the context length, leading us to perform numerous expensive encoder passes. Furthermore, we cannot store these encoder outputs when they were first collected - since the encoder’s parameters are changing as we update our model, this means the encoder outputs will be different, which can lead to performance degradation if the encoder has changed significantly. This topic was explored in detail in R2D2 [1].
>
> **Question 2 (Use of the current observation):** Please see equation 1; we do use the pretrained vision encoder on the current observation ($o_t$ is used twice, one for each encoder). The encoding of the current observation is the final encoding that is passed through the LSTM.
>
> **Conclusion:** We have addressed all except one of your concerns with rigor. While we cannot apply RISE to Dreamer-v3 or MEME, we have provided detailed and justifiable reasoning why this is infeasible. Given this response, we hope you are willing to reconsider your score.
>
> **References:**
>
> [1] Kapturowski, Steven, et al. "Recurrent experience replay in distributed reinforcement learning." International conference on learning representations. 2018.

---

### Official Review · Reviewer_yJbh · 2025-11-03

**Soundness:** 2
**Presentation:** 3
**Contribution:** 2
**Rating:** 6
**Confidence:** 4

**Summary:**

This paper proposes a new approach for recurrent reinforcement learning that reduces computational overheads leveraging non-learnable encoder (pre-trained model). They use an attention style mechanism to fuse the LSTM stream (encoded interaction history) and the CNN stream (encoded current state). They provide experimental analysis across different benchmarks and highlight their computational gain.

**Strengths:**

### Strengths:

1. This work proposes a new framework that requires a single pass of the observation with the current learnable model and uses embeddings from pre-trained vision models for previous state sequence. This significantly reduces the computational burden.

2. Experiments are conducted across four different benchmarks. Experimental results are encouraging to an extent, especially in terms of computational overhead.

3. While their approach introduces a number of design choices, they provide thorough analysis of different possible choices. It is evident that the performance is not highly sensitive to those choices.

**Weaknesses:**

### Weaknesses:

1. While R2D2 has been mentioned as a similar prior work, no comparison with R2D2 has been presented. Further, comparison with other recent transformer-based approaches would be great to assess the wider impact of the work.

2. The performance improvement is limited to a subset of environments within a benchmark. Such as in Procgen or Vizdoom, a certain number of environments see reward gain while using RISE.

**Questions:**

1. How did you obtain the human-normalized score for Procgen? Does not the original paper only provide reference for PPO-normalization and min-max normalization?

---

> ### Author Response · Authors · 2025-11-14
>
> Thank you for your thoughtful review and for recognizing our contribution of significantly reducing computational burden, detailed evaluation, and thorough analysis. We will now systematically discuss each of the outstanding issues:
>
> **Weakness 1 (R2D2 Comparison):** Were you hoping to see R2D2 applied to the BTR algorithm, or a comparison of BTR + RISE against R2D2? If you are hoping for the former, we are happy to apply R2D2’s style to BTR and report our results on Atari-5 with 3 seeds before the end of the rebuttal period. We will do this by matching the effective batch sizes for a fair comparison: BTR + RISE updates a total of 256 states per batch (with one encoder pass per state), thus we can use a similar budget for R2D2 by using a batch size of 3, and a context length of 86, resulting in performing updates on 86*3=258 different states, and performing 258 encoder passes. This should provide a fair walltime-balanced comparison between the two approaches. If you were referring to comparing BTR+RISE’s results against the R2D2 algorithm, we would argue this is an unfair comparison since R2D2 uses 90 billion training frames compared to BTR + RISE’s 200 million.
>
> **Weakness 2 (Selective Performance Improvement):** As explained in lines 257-258, we believe this is likely due to some environments not having meaningful partial observability or not having long-term credit assignment problems. Furthermore, RL components only helping in some environments within a benchmark has been an extremely common trend, with almost no single component causing improvement in all environments. Please see Rainbow DQN’s [1] Figure 4 for a convincing example, including 6 different example components. Please see our general comment for more information.
>
> **Question 1 (Procgen Normalization):** We did not use human-normalized scores for procgen, but rather used the min-max normalized as per the original paper. Figure 6 and 23 do state that we used min-max normalization; however, Figure 22 does say human-normalized: this was a typo, which we thank you for noticing, and will promptly fix.
>
> **Conclusion:** We will reply again with the results of BTR using R2D2-style training, under the settings we described. We believe this meticulously addresses all of your concerns, and we hope this leads you to advocate for acceptance and a raising your score.
>
> **References:**
>
> [1] Hessel, Matteo, et al. "Rainbow: Combining improvements in deep reinforcement learning." Proceedings of the AAAI conference on artificial intelligence. Vol. 32. No. 1. 2018.

---

### Author Response · Authors · 2025-11-14

We would like to thank all reviewers for their constructive review comments. We found that all reviewers agree that our work identifies an important problem - off-policy recurrent RL has a high computational burden which prevents the majority of researchers and practitioners from using such algorithms. Additionally, our work produces a plug-and-play solution that is effective and simple. Furthermore, reviewers appeared to particularly value our thorough evaluation, including 88 environments from 4 different benchmarks.

We would also like to take this opportunity to thoroughly and rigorously address two points that may impact the decisions of several reviewers:

**Selective Environment Performance Improvement:** It was pointed out that RISE tended to improve performance in specific environments, rather than all environments across different benchmarks. In lines 256-257, we acknowledge that RISE does not improve performance across environments, and nor should it. Our work focuses on allowing recurrent models to be cheaply used - while many environments are improved due to handling partial observability or better long-term credit assignment, not all environments will include these (as mentioned in lines 257-258). For example, environments with hard exploration where no rewards are seen (such as Atari Montezuma’s Revenge, or Miniworld’s ThreeRooms) would not be expected to improve performance. Furthermore, BTR already reaches near-optimal performance on many environments (such as Atari Boxing, Miniworld’s WallGap or TMaze), meaning there is no way to achieve performance improvement. Lastly, selective improvement is remarkably common among RL algorithms - please see Rainbow DQN’s Figure 4 [1], which demonstrates that when removing 6 different components from Rainbow DQN, almost all of them had varying effects across different environments. We believe this demonstrates that this is not a significant weakness of our work.

**Additional Experiments:** Many reviewers suggested additional experiments that would strengthen our work:
- Deeper exploration of design decisions - since RISE uses the two-encoder architecture, leaves the question of how to fuse to two streams. To do a more detailed analysis of these decisions, we will add at least 3 additional experiments. Firstly, rather than multiplying the two streams together, adding them is another viable option. Secondly, while we upscale the LSTM’s output such that the two streams can be multiplied, another option is to simply concatenate the output of the LSTM with the learnable encoder’s output. Lastly, It is possible to combine the two streams before the LSTM without substantial cost -  we will test a variant which multiplies the learnable encoder’s downsampled output with the non-learnable encoder’s output.
- Comparison of BTR with R2D2’s style. While BTR typically only updates a single Q-value per batch item, it is possible to adopt an R2D2 style, whereby temporally correlated sequences of transitions are updated. While this is typically done with long sequences and large batches, resulting in heavy-compute, we will test a compute-balanced version of this, using a batch size of 3, with sequences of 85, resulting in 258 total Q-values being updated, and 258 total encoder passes per gradient step.
- Lastly, it was suggested by Reviewer Wzuf that we could try replacing the non-learnable encoder with a slow-moving EMA of the learnable encoder. This has the benefit of being more adaptable, but the disadvantage of not having a pre-trained model as a prior, and introducing stale embeddings in the replay buffer. We think this is an interesting experiment idea, which would add value to the paper.

We hope that this clarification and additional experiments will further strengthen our paper. Before the end of the rebuttal period, we will upload our revised PDF.

[1] Hessel, Matteo, et al. "Rainbow: Combining improvements in deep reinforcement learning." Proceedings of the AAAI conference on artificial intelligence. Vol. 32. No. 1. 2018.

---

### Author Response · Authors · 2025-11-23
**Paper Rebuttal Revision**

We have now uploaded our revised PDF, including numerous changes to improve the quality of the paper and rigorously address any remaining concerns. We performed 8 new experiments with 3 seeds each on Atari-5 benchmark to strengthen our analysis. Given that these experiments add a very significant amount of information and rigor to the paper, we hope that reviewers will be willing to raise their scores. Please find a list of the changes below.

 - Section 5.1 now includes ablations of using both the main encoder, and an exponential moving average (EMA) of the main encoder for the RISE encoding.
 - Added new analysis subsection 5.2, which ablates different ways to combine the two streams, including addition, concatenation and combining the streams before the LSTM.
- Added new analysis subsection 5.5, where we perform a detailed analysis of RISE vs R2D2, finding that RISE substantially outperforms - R2D2 when using remotely similar amounts of compute. We also found that BTR + R2D2 performed very poorly when using small batches, indicating that preventing temporally correlated batches is very important (which RISE prevents without the need for excessive compute).
- Section 5.1 specifies that RISE’s memory usage does not depend on the context length.
- In Section 5.1, we include a discussion on the negligibly increased memory/GPU memory usage.
- Fix typo on line 472 “For example, RISE could be used with transformers for image-based text-based tasks, or CNNs for audio-based tasks”
- In Section 7, added the sentence “We also acknowledge that the use of a fixed, pre-trained encoder may limit the model’s ability to extract useful features in some environments. In this sense, RISE acts as a trade-off, allowing greater compute-efficiency in exchange for less adaptability if the encoder does not capture the relevant features.”, to acknowledge this limitation.
- Figure 26 (Procgen’s IQM performance figure) fixed a typo where it said “Human-Normalized Performance”, instead of “Min-Max Normalized Performance”
- Appendix B now includes individual performance graphs for all of our new experiments from sections 5.1, 5.2 and 5.5.

---

### Comment · Area_Chair_fvf8 · 2025-11-26

Dear Reviewers,

Thank you for sharing your valuable insights and expertise, which have played an important role in the review process.

In response to the initial feedback, the authors have submitted a detailed rebuttal addressing the comments raised by the reviewers.

I would appreciate it if you could carefully review their response and consider how it may affect your initial evaluation.

Please feel free to share your updated thoughts or any additional comments after reviewing the rebuttal.

Thank you again for your time and contributions.

---

### Meta-Review · Area_Chair_VMP8 · 2026-01-06

**Summary:**

The paper proposes RISE, a method to reduce computational costs in recurrent off-policy RL by using fixed pretrained encoders for historical observations. Reviewers acknowledge the practical importance of reducing computational burden but raise concerns about limited novelty (primarily an engineering optimization rather than new RL principles), incomplete analysis of when fixed encoders may fail, and selective performance improvements across environments [Reviewers Wzuf, VpvB, yJbh]. The rebuttal provided additional ablations but did not fully address theoretical gaps or demonstrate broader applicability beyond visual benchmarks.

**Reviewer Concerns:**

Addressed by rebuttal: The authors provided requested experiments comparing RISE to R2D2-style training and EMA-based encoders, demonstrating RISE's computational advantages [Reviewer yJbh]. They clarified the dual-stream architecture integration mechanism and added ablations on fusion methods (addition, concatenation, pre-LSTM combination) [Reviewer dZ14].

Outstanding concerns: The core criticism that this is primarily an engineering contribution ("making an existing recipe cheaper") rather than a novel RL principle remains unaddressed [Reviewer Wzuf]. The theoretical analysis of why fixed encoders preserve credit assignment is still underdeveloped [Reviewer dZ14]. Concerns about generalization to non-visual domains and out-of-distribution tasks persist [Reviewers WuVd, Wzuf].

**Reviewer Scores:**

Reviewer Wzuf: 2. The reviewer may maintain their score as the fundamental concern about limited conceptual novelty—viewing this as systems optimization rather than new RL methodology—was not substantively addressed despite additional experiments.

Reviewer VpvB: 4. The reviewer may maintain their score as the inability to apply RISE to stronger baselines (MEME, Dreamer-v3) and limited theoretical contribution remain unresolved.

Reviewer yJbh: 6. The reviewer may maintain their score as the R2D2 comparison was provided, though selective environment improvements remain a concern.

Reviewer WuVd: 6. The reviewer may maintain their score as responses were helpful but did not significantly change their assessment regarding theoretical limitations.

Reviewer dZ14: 6. The reviewer explicitly stated they would keep their original rating despite appreciating the authors' response.

---

### Decision · Program_Chairs · 2026-01-26

Reject